# Fibromodulin selectively accelerates myofibroblast apoptosis in cutaneous wounds by enhancing interleukin 1β signaling

Wenlu Jiang[1,9], Xiaoxiao Pang[2,9], Pin Ha ®[1], Chenshuang Li[3], Grace Xinlian Chang[1], Yuxin Zhang[2], Lawrence A. Bossong[4], Kang Ting ®[5,6] ✉, Chia Soo ®[1,7] ✉ & Zhong Zheng ®[1,8] ✉

Activated myofibroblasts deposit extracellular matrix material to facilitate rapid wound closure that can heal scarlessly during fetal development. However, adult myofibroblasts exhibit a relatively long life and persistent function, resulting in scarring. Thus, understanding how fetal and adult tissue regeneration differs may serve to identify factors that promote more optimal wound healing in adults with little or less scarring. We previously found that matricellular proteoglycan fibromodulin is one such factor promoting more optimal repair, but the underlying molecular and cellular mechanisms for these effects have not been fully elucidated. Here, we find that fibromodulin induces myofibroblast apoptosis after wound closure to reduce scarring in small and large animal models. Mechanistically, fibromodulin accelerates and prolongs the formation of the interleukin 1β-interleukin 1 receptor type 1-interleukin 1 receptor accessory protein ternary complex to increase the apoptosis of myofibroblasts and keloid- and hypertrophic scar-derived cells. As the persistence of myofibroblasts during tissue regeneration is a key cause of fibrosis in most organs, fibromodulin represents a promising, broad-spectrum antifibrotic therapeutic.

Essential for injured tissue regeneration, wound healing involves several interrelated dynamic phases with overlapping time courses. In particular, granulation tissue formation and contraction are critical for reestablishing tissue continuity and reducing wound size. In granulation tissue, cytokines and mechanical stress activate fibroblasts, converting them into myofibroblasts. These activated myofibroblasts synthesize and deposit extracellular matrix (ECM) material to replace the provisional fibrin-rich matrix formed in the hemostasis phase of wound healing. Myofibroblasts also express α-smooth muscle actin (α-SMA) in microfilament bundles or stress fibers and acquire a contractile

[1]Division of Plastic and Reconstructive Surgery, David Geffen School of Medicine, University of California, Los Angeles, Los Angeles, CA 90095, USA. [2]Chongqing Key Laboratory of Oral Diseases and Biomedical Sciences, Chongqing Municipal Key Laboratory of Oral, Biomedical Engineering of Higher Education, Stomatological Hospital of Chongqing Medical University, Chongqing 401147, China. [3]Department of Orthodontics, School of Dental Medicine, University of Pennsylvania, Philadelphia, PA 19104, USA. [4]Department of Neuroscience, Princeton University, Princeton, NJ 08540, USA. [5]American Dental Association Forsyth Institute, Cambridge, MA 02142, USA. [6]School of Dentistry, National Yang-Ming Chiao Tung University, Taipei 30010, Taiwan. [7]Department of Orthopaedic Surgery and the Orthopaedic Hospital Research Center, University of California, Los Angeles, Los Angeles, CA 90095, USA. [8]Department of Periodontics, School of Dental Medicine, University of Pennsylvania, Philadelphia, PA 19104, USA. [9]These authors contributed equally: Wenlu Jiang, Xiaoxiao Pang. ✉e-mail: erickangting@gmail.com; bsoo@g.ucla.edu; leozz95@gmail.com

phenotype that reduces the wound area to promote closure[1,2]. Unlike their rapid disappearance from fetal wounds that heal scarlessly[2-5], myofibroblasts in adult wounds are relatively long-lived, sustaining a contractile force over prolonged periods and synthesizing and depositing excessive, dysfunctional ECM, leading to permanent scarring[6]. Although myofibroblast apoptosis occurs upon the reestablishment of tissue integrity to minimize excessive contraction under normal circumstances[1,2,6], wounded tissue functionality is not wholly restored in adults due to scar formation. Under pathological conditions, myofibroblasts can escape from apoptosis to continuously function, resulting in deposition and accumulation of contracted collagen fibers that elevate the skin surface and increase tissue stiffness, becoming the ultimate culprits responsible for keloid and hypertrophic scars (HSs)[4,7-11].

Despite numerous development attempts, no clinical drugs exist today that reduce or prevent dermal fibrosis or scarring[1,2], and why some people are more prone to keloid and HS formation is not fully elucidated. The complexity of wound healing management can be partially attributed to the dual role of myofibroblasts. In the early healing phase, myofibroblast contraction is indispensable for wound closure, rendering any strategies that inhibit myofibroblast activation impractical. In contrast, during the later phase, myofibroblast persistence causes fibrogenesis, necessitating selective myofibroblast elimination strategies that are difficult to achieve due to the lack of myofibroblast-specific surface marker(s). Thus, understanding the dynamic equilibrium between the activation and elimination of myofibroblasts is paramount to optimal repair with less scarring. Notably, unlike adult wounds that heal with scarring, fetal wounds heal scarlessly[8] and display a rapid myofibroblast activation and deactivation profile, whereby activated fetal myofibroblasts are short-lived due to accelerated apoptosis[5,12]. These observations provide valuable insight into the idea that emulating a fetal-like pattern of quick myofibroblast activation followed by expedited clearance could be optimal for reducing scarring in adult wounds.

Our previous studies demonstrate that fibromodulin (FMOD), a small leucine-rich proteoglycan (SLRP), is required for fetal-type scarless repair[8]. Exogenous FMOD administration significantly improved gross scar appearance, reduced scar size, and accelerated the reestablishment of normal tensile tissue strength in small and large animal models that simulate normal and hypertrophic adult human wound repair[13,14]. Furthermore, our previous studies demonstrated that FMOD has a striking capability to elicit fetal-like characteristics in adult fibroblasts[13], and adult dermal fibroblasts cultured under long-term FMOD exposure can even acquire tri-lineage differentiation potency[15-18]. Specifically, FMOD selectively prolongs canonical, Smad-driven transforming growth factor (TGF)β1 signaling to enhance fibroblast migration and accelerate myofibroblast conversion, activation, and contraction to promote timely wound closure while simultaneously attenuating non-Smad-driven TGFβ1 signaling to reduce TGFβ1 autoinduction and downstream fibrotic factor expression for less scar formation[13,19]. Having delineated the influence of FMOD on adult fibroblasts and wounds during the early phase repair, we set out to investigate FMOD's effects on adult myofibroblasts—the critical player in healing and scarring—at the later healing phase.

Here, we show that FMOD enhances and extends the assembly of the ternary complex consisting of interleukin (IL)1β, interleukin 1 receptor type 1 (IL1R1), and interleukin 1 receptor accessory protein (IL1RAP), thereby promoting another fetal-like attribute of myofibroblasts, namely, their accelerated apoptosis and rapid clearance following wound closure.

## Results

### FMOD in closed cutaneous wounds leads to rapid myofibroblast clearance

We first used gain- and loss-of-function models to determine the effect of FMOD on myofibroblasts. Adult FMOD-knockout (Fmod[-/-]) mice

exhibited delayed reepithelialization[20], impaired angiogenesis[21], and larger scar size[13], along with considerably higher granulation tissue α-SMA expression, than wild-type (WT) mice (Supplementary Fig. 1). Intradermal FMOD injection markedly reduced the density of α-SMA⁺ myofibroblasts in both WT and Fmod[-/-] mouse wounds (Supplementary Fig. 1).

Noticeably, in addition to reducing the scar size[13], FMOD markedly decreased the population of active myofibroblasts in adult rat wounds (Fig. 1a, b), consistent with our previous data showing downregulation of Acta2 (the gene encoding α-SMA) expression in closed, FMOD-treated skin wounds[13]. Meanwhile, FMOD-treated wounds have a higher percentage of cells stained with cleaved-caspase-3 (Fig. 1a and c), an indicator of programmed cell death[22].

Furthermore, in an established excessive-mechanical-loading adult female red Duroc porcine wound model, which most closely approximates HS formation in humans[23], FMOD significantly reduced scar size, improved gross wound appearance[14], decreased myofibroblast persistence, and increased percentage of cleaved-caspase-3⁺ cells in the remodeling wounds (Fig. 1d–f).

### FMOD selectively induces myofibroblast apoptosis

Having established an association between wound myofibroblasts and the presence or absence of FMOD, we next explored how FMOD impacts myofibroblast programmed cell death. Firstly, TGFβ1, a central moderator of myofibroblast activation and survival[10,24], was employed to stimulate the conversion of rat dermal fibroblasts (RDFs) into activated myofibroblast derivatives in vitro (Supplementary Fig. 2a). As a positive control, IL1β, a key mediator of myofibroblast apoptosis, selectively induced the apoptosis of RDF-converted myofibroblasts (RDF-myofibroblasts), but not unconverted RDFs (Supplementary Fig. 2b, c), consistent with previous observation in lung and corneal tissues[25,26]. Meanwhile, TGFβ1 rescued RDF-myofibroblasts from IL1β-induced apoptosis. FMOD, like IL1β, selectively stimulated the apoptosis of RDF-myofibroblasts but not unconverted RDFs; however, FMOD-induced apoptosis of RDF-myofibroblasts was not inhibited by TGFβ1 (Supplementary Fig. 2c).

Next, we examined whether FMOD-induced myofibroblast apoptosis extends to those derived from normal human dermal fibroblasts (NHDFs). To ensure high reproducibility, we first used commercially available neonatal foreskin-derived BJ fibroblasts. By flow cytometry, we found that post-activation with TGFβ1 (Fig. 2a), approximately 66% of BJ fibroblasts had converted into myofibroblasts (BJ-myofibroblasts) by day 6 (Fig. 2b, c), with a significant upregulation of ACTA2 expression (Fig. 2d). Mirroring the results with RDF-myofibroblasts, FMOD was as effective as IL1β in triggering BJ-myofibroblast apoptosis (identified as the Annexin V⁺/propidium iodide (PI)⁻ cell population by flow cytometry), even in the presence of TGFβ1 (Fig. 2e, f). Conversely, FMOD did not induce apoptosis of unconverted BJ fibroblasts (Supplementary Fig. 3).

Recent studies indicated that caspase-3 is also involved in cell pyroptosis[27,28], another form of programmed cell death. However, the pyroptotic BJ-myofibroblasts (identified as the Annexin V⁺/PI⁺ population by flow cytometry) did not increase in response to FMOD treatment (Fig. 2e, g), revealing that FMOD primarily promotes apoptosis, rather than pyroptosis, in myofibroblasts.

To further validate the effect of FMOD on myofibroblast apoptosis, we used TGFβ1 to convert three adult NHDFs (FB-AA36, FB-C31, and FB-C51, sourced from donors of different races, ages, and genders; Supplementary Table 1) into myofibroblasts. IL1β and FMOD induced apoptosis of the activated NHDF-myofibroblasts to a similar degree (Fig. 3 and Supplementary Fig. 4), while neither agent promoted the apoptosis of unconverted NHDFs. These findings confirm that FMOD induces targeted apoptosis of myofibroblasts but not unconverted fibroblasts.

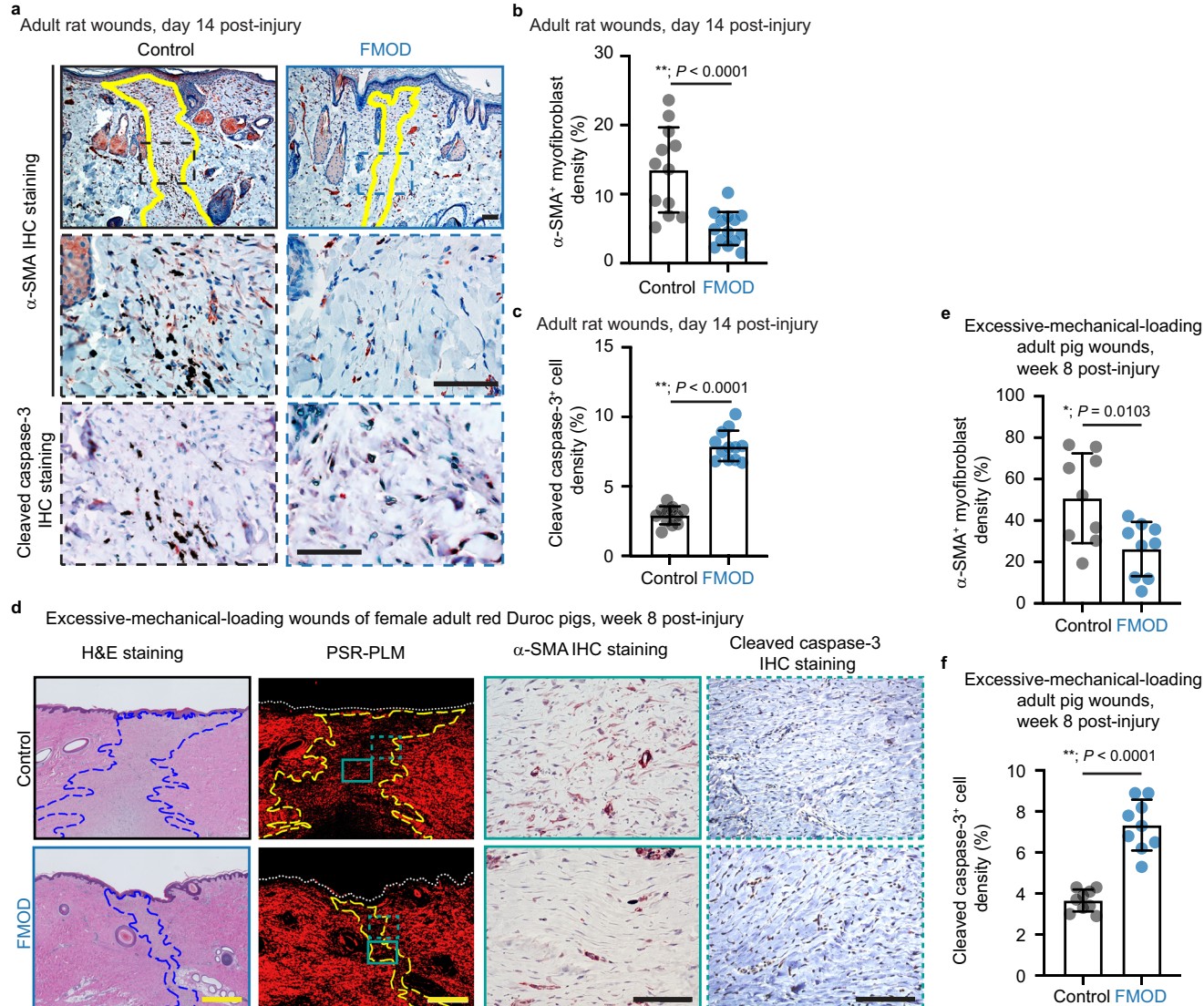

**Fig. 1 | Fibromodulin (FMOD) accelerates myofibroblast clearance in rat and pig models with reduced scar formation. a** Representative images of sections stained with α-smooth muscle actin (α-SMA) and cleaved caspase-3 by immunohistochemistry (IHC) staining from adult rat wounds at day 14 post-injury. Yellow lines outline the scar area. Dashed boxes in the lower magnification images represent the region of interest shown in the higher magnification images below. Scale bars, 100 μm. **b** Quantification of α-SMA⁺ myofibroblast density in adult rat wounds from (**a**). N = 12 (control) or 14 (FMOD)-treated rats, respectively. **c** Quantification of cleaved caspase-3⁺ cell density in adult rat wounds from (**a**). N = 12 rats. **d** Representative images of sections stained with hematoxylin & eosin (H&E) (first panel), picrosirius red (PSR) coupled with polarized light microscopy (PLM) (second panel), IHC for α-SMA (third panel) and cleaved caspase-3 (fourth panel) from excessive-mechanical-loading wounds of female adult red Duroc pigs at week 8 post-injury. Dermal scar areas are outlined in blue (for the H&E-stained image) or yellow (for PSR-PLM images), while white dotted lines in the PSR-PLM images outline the epidermal edge. The solid green-boxed regions (second panel) represent the region of α-SMA-stained images, while the dotted green-boxed regions (second panel) represent the region of cleaved caspase-3-stained images. Scale bars, 50 (black) or 500 μm (yellow). **e** Quantification of α-SMA⁺ myofibroblast density in control- vs. FMOD-treated adult pigs from **d**. **f** Quantification of cleaved caspase-3⁺ cell density in control- vs. FMOD-treated adult pigs from (**d**). N = 9 wounds from 3 pigs (**e, f**). All treatments were administered at the time of surgery. The number of IHC-positively stained cells and nuclei across the entire wound area was counted under a microscope from two centrally bisected sections of each wound sample. The ratio of IHC-positively stained cells to the total number of cells (indicated by the number of nuclei) was then calculated to quantify the density of α−SMA⁺ (**b, e**) or cleaved caspase-3⁺ (**c** and **f**) cells. Data presented as mean ± standard deviation (s.d.) overlaying all the data points. P-values were determined by two-tailed unpaired t-tests (**b–f**). *P < 0.05; **P < 0.005. Source data are provided as a Source Data file.

## FMOD induces apoptosis in the keloid- and HS-derived fibroblasts

We subsequently investigated the impact of FMOD on keloid- and HS-derived fibroblasts. Given the absence of a suitable animal model for keloids[29], we first used a commercially available human keloid-derived fibroblast cell line, KB-AA35. KB-AA35 cells were obtained from a 35-year-old African−American female donor who was race-, gender- and age-matched to the donor of NHDF FB-AA36 (Supplementary Table 1). KB-AA35 cells exhibited a significantly higher *ACTA2* expression at baseline compared to FB-AA36 cells, while a subset of KB-AA35 cells exhibited notably strong α-SMA staining (Fig. 4a, b). At the same time, FB-AA36 (and other unconverted NHDFs) did not display α-SMA signals (Fig. 3a and Supplementary Fig. 4a, e). In addition, TGFβ1 induced significantly more intense α-SMA expression in KB-AA35 cells than in FB-AA36 cells (Supplementary Fig. 5). Interestingly, a lower TGFβ1 concentration that was not effective in converting FB-AA36 cells to myofibroblasts by day 6, strongly induced α-SMA expression in KB-AA35 cells as early as day 3 (Supplementary Fig. 5). Importantly, both

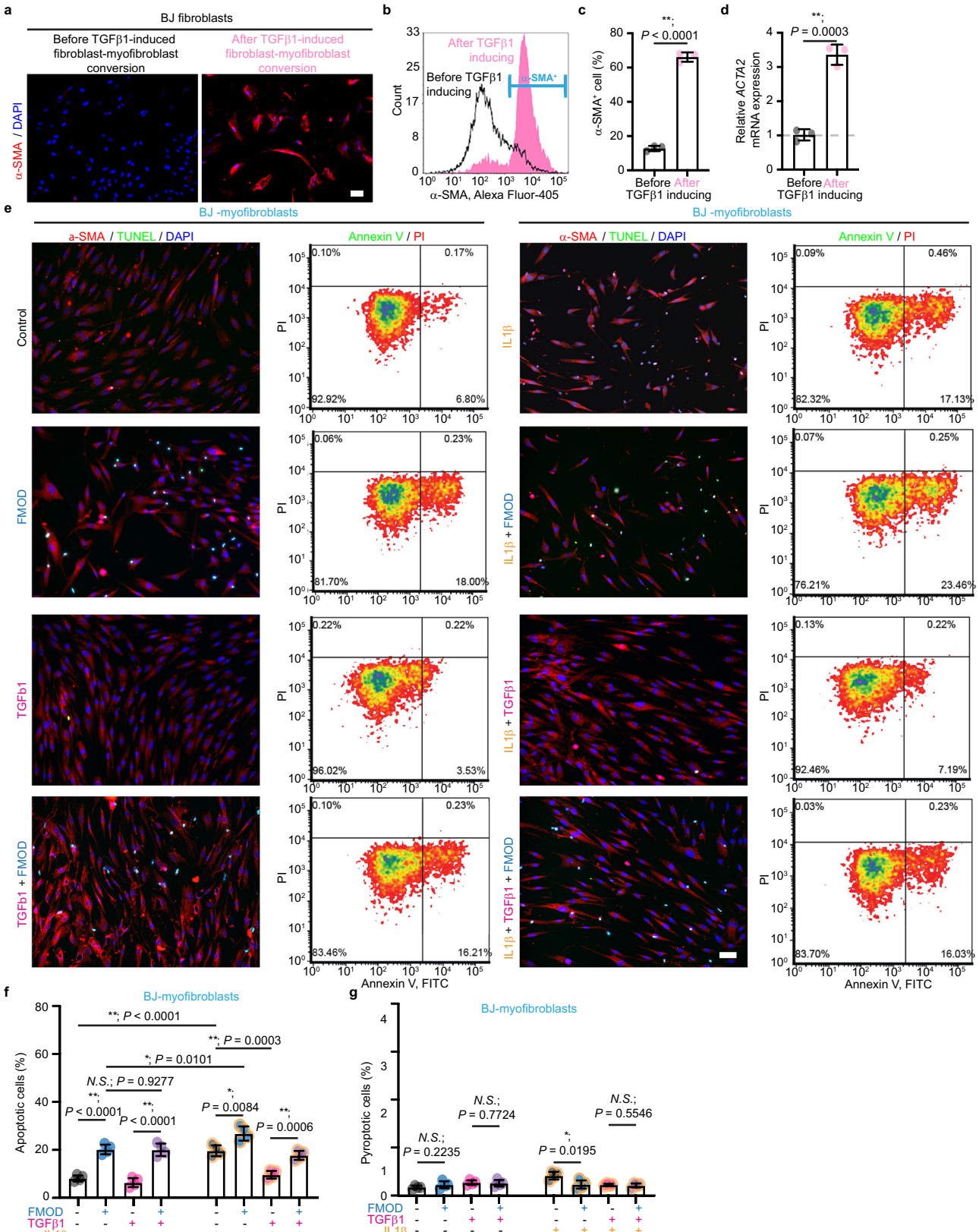

IL1β and FMOD induced apoptosis in the KB-AA35 cells to a similar degree, while the concomitant presence of TGFβ1 decreased IL1β-mediated apoptosis, but not FMOD-mediated apoptosis (Fig. 4c).

To confirm the effect of FMOD on human scar fibroblasts, we sourced normal skin- and HS-derived dermal fibroblasts from the same donor and location via the National Disease Research Interchange

(NDRI). One sample comprised normal skin and keloid tissue from the abdomen, a region of low mechanical loading (Donor reference ID 1914138; Supplementary Table 1). In line with the previous observations[9], a subset of abdomen keloid fibroblasts displayed intensive α-SMA staining, with α-SMA staining predominately observed around microvessels (Fig. 5a). Alike KB-AA35 cells, keloid-

**Fig. 2 | Fibromodulin (FMOD) induces BJ fibroblast-converted myofibroblast apoptosis.** All cells were subjected to serum starvation prior to treatment. **a** Representative images of human BJ fibroblasts stained with α-smooth muscle actin (α-SMA) by immunofluorescence staining before and after 5.0 ng/mL transforming growth factor (TGF)β1 treatment for 6 days. $N = 3$ biological replicates. **b** Representative flow cytometry plot of BJ fibroblasts before and after 5.0 ng/mL TGFβ1 treatment for 6 days, with mouse anti-α-SMA antibody [4A4] (GTX60466, GeneTex; 1: 250 dilution) and goat anti-mouse IgG(H + L) highly cross-adsorbed secondary antibody, Alexa Fluor™ 405 (A-31553, Thermo Fisher Scientific; 2 μg/mL). **c** Quantification of α-SMA⁺ myofibroblast percentages in BJ fibroblasts from (**b**). $N = 3$ biological replicates. **d** Expression of *ACTA2* (the gene encoding α-SMA) in BJ fibroblasts before and after fibroblast-myofibroblast conversion. Data were normalized to the *ACTA2* level before TGFβ1-induced fibroblast-myofibroblast conversion. $N = 3$ biological replicates. **e** Representative images of BJ fibroblast-converted myofibroblasts (BJ-myofibroblasts) with α-SMA and TUNEL staining, accompanied by the respective flow cytometry plots stained with Annexin V-FITC and PI staining. $N = 3$ (for α-SMA and TUNEL staining) or 4 (for flow cytometry) biological replicates, respectively. **f** Quantification of apoptotic BJ-myofibroblasts by flow cytometry from (**e**). **g** Quantification of pyroptotic BJ-myofibroblasts by flow cytometry from (**e**). $N = 4$ biological replicates (**f, g**). Scale bars, 25 μm. Data presented as mean ± s.d. overlaying all the data points. *P*-values were determined by two-tailed unpaired *t*-tests (**c**–**g**). *N.S.*, not significant, $P > 0.05$; *$P < 0.05$; **$P < 0.005$. Source data are provided as a Source Data file.

derived fibroblasts exhibited a higher *ACTA2* expression at baseline than normal skin-derived fibroblasts (Fig. 5b). Moreover, akin to KB-AA35 cells, both IL1β and FMOD induced apoptosis in these abdomen keloid-derived dermal fibroblasts but not the normal skin-derived fibroblasts (Fig. 5c–e). Another donor (Donor reference ID 2106181) with keloid and normal tissue collected from the right knee, a region of higher mechanical loading, also exhibited the same dermal fibroblast apoptosis profile (Supplementary Fig. 6), further validating our observations.

In addition to keloid abdominal tissue, donor 1914138 also provided HS tissue from the left shoulder, an area of high mechanical loading (Supplementary Table 1). The HS tissue of donor 1914138 also exhibited a higher α-SMA expression than the normal skin, and both IL1β- and FMOD-induced HS-derived fibroblast apoptosis (Fig. 5). To verify that the apoptosis observed in HS-derived dermal fibroblasts was not due to variations in body location, we procured two additional pairs of normal skin and HS tissues (Supplementary Table 1), one from the abdomen (from a donor with reference ID 1914476, representing a low mechanical-loading region) and the other from the left elbow (from a donor with reference ID 1914478, representing a high mechanical-loading area). These HS tissues and their dermal-derived fibroblasts exhibited higher α-SMA expression than the donor-matched, location-matched normal skin. Furthermore, IL1β and FMOD enhanced apoptosis in these HS-derived fibroblasts (Supplementary Fig. 7). These data conclusively demonstrate that FMOD induces apoptosis in at least a subset of keloid- and HS-derived fibroblasts.

#### FMOD-induced myofibroblast apoptosis requires IL1β signaling

Interestingly, despite observing higher baseline *IL1B* transcription levels in NHDFs compared to their myofibroblast derivatives, an enzyme-linked immunoassay (ELISA) conducted on the culture medium of fibroblasts or activated myofibroblasts failed to detect any active IL1β without FMOD stimulation (Supplementary Fig. 8). Concomitantly, FMOD only promoted IL1β mRNA and protein expression of activated myofibroblasts, but not their unconverted NHDF precursors (Supplementary Fig. 8). However, FMOD was capable of promoting IL1β expression in keloid- and HS-derived fibroblasts (Fig. 4d, e and Supplementary Fig. 9).

As our data suggested that exogenous IL1β is not required for FMOD to exert its pro-apoptosis potency on myofibroblasts, we utilized the CRISPR/Cas9 platform[30,31] to fully knock out endogenous *IL1B* expression in BJ fibroblasts to definitively determine if FMOD-mediated apoptosis requires endogenous IL1β. The knockout was confirmed by the inability of lipopolysaccharide to induce active IL1β secretion from *IL1B*⁻/⁻ BJ fibroblasts (Supplementary Fig. 10a). Importantly, deletion of *IL1B* did not prevent TGFβ1-induced fibroblast-to-myofibroblast conversion (Supplementary Fig. 10b, c). Although *IL1B*⁻/⁻ BJ-myofibroblasts displayed a lower apoptosis baseline than WT BJ-myofibroblasts, exogenous IL1β induced a similar level of apoptosis in WT and *IL1B*⁻/⁻ BJ-myofibroblasts (Fig. 6a, b), indicating that deletion of *IL1B* did not diminish the apoptotic potential of BJ-myofibroblasts (Fig. 6c). However, deletion of *IL1B* in BJ-myofibroblasts eliminated

FMOD's ability to induce myofibroblast apoptosis (Fig. 6) and active IL1β production (Supplementary Fig. 10d), delineating the essential role of endogenous IL1β in FMOD-induced myofibroblast apoptosis.

To further elucidate the interplay between FMOD and IL1β signaling in myofibroblast apoptosis, two IL1β signaling inhibitors, rhein and an IL1 receptor antagonist (IL-1Ra), were used. Rhein inhibits the IL-1 converting enzyme (ICE; *aka.*, caspase-1, a highly selective protease cleaving the intracellular IL-1β inactive precursor) from releasing active IL-1β[32–34], and IL-1Ra competes with IL1β for binding to the cell surface IL1R1, thereby preventing IL1β intracellular signaling[35]. We found that rhein effectively blocked FMOD-induced active IL1β production and targeted apoptosis of BJ-myofibroblasts (Supplementary Fig. 11). Meanwhile, the stimulation of FMOD on BJ-myofibroblast IL1β expression and apoptosis was also abolished by IL-1Ra (Supplementary Fig. 12), which is consistent with the results in RDF-myofibroblasts (Supplementary Fig. 13). Together, these data demonstrate that endogenous IL1β signaling is essential for FMOD-mediated myofibroblast apoptosis.

#### FMOD enhances and prolongs IL1β-IL1R1-IL1RAP ternary complex formation on the surface of myofibroblasts

To identify the essential components of IL1β signaling for FMOD-mediated myofibroblast apoptosis, we employed RNA interference (RNAi) to knock down the expression of *IL1B*, *IL1R1*, and *IL1RAP* in BJ-myofibroblasts (Supplementary Fig. 14). Notably, IL1R1 is the cognate receptor and IL1RAP is the co-receptor for IL1β[36]. All the knockdown BJ-myofibroblasts exhibited the same baseline IL1β production levels as the control BJ-myofibroblasts transfected with non-targeting siRNAs (siRNA-control). However, FMOD only increased IL1β secretion in the siRNA-control cells but not any of the three knockdown BJ-myofibroblasts (Fig. 7a, b). As expected, the knockdown of *IL1B*, *IL1R1*, and *IL1RAP* notably reduced FMOD's capacity to trigger apoptosis in BJ-myofibroblasts (Fig. 7c–e), indicating the assembly of adequate IL1β-IL1R1-IL1RAP ternary complex is a prerequisite for induction of myofibroblast apoptosis mediated by FMOD. Interestingly, all myofibroblasts derived from BJ fibroblasts that underwent siRNA transfection presented a higher percentage of pyroptotic cells (Fig. 7c and f), which is aligned with recent findings that noncoding RNAs enhance cell pyroptosis[37,38]. However, BJ-myofibroblasts, even when subjected to RNAi, did not exhibit an increase in pyroptosis after FMOD treatment (Fig. 7g), reinforcing our conclusion that FMOD induces myofibroblast apoptosis but not pyroptosis.

Next, binding studies revealed that IL1R1 and IL1RAP, but not IL1β, directly bound FMOD (Fig. 8a–c). The pull-down assay further demonstrated binding between FMOD and the IL1β-IL1R1-IL1RAP ternary complex in an acellular system (Fig. 8d). To validate FMOD's binding to the IL1β-IL1R1-IL1RAP ternary complex under a biological condition, we extracted membrane proteins from BJ-myofibroblasts. To account for potentially anomalous migration in SDS-PAGE (i.e., 'gel shifting') of membrane proteins[39], we first used Western blotting to individually identify membrane-associated IL1R1 and IL1RAP as reference points (Supplementary Fig. 15). We then confirmed that FMOD

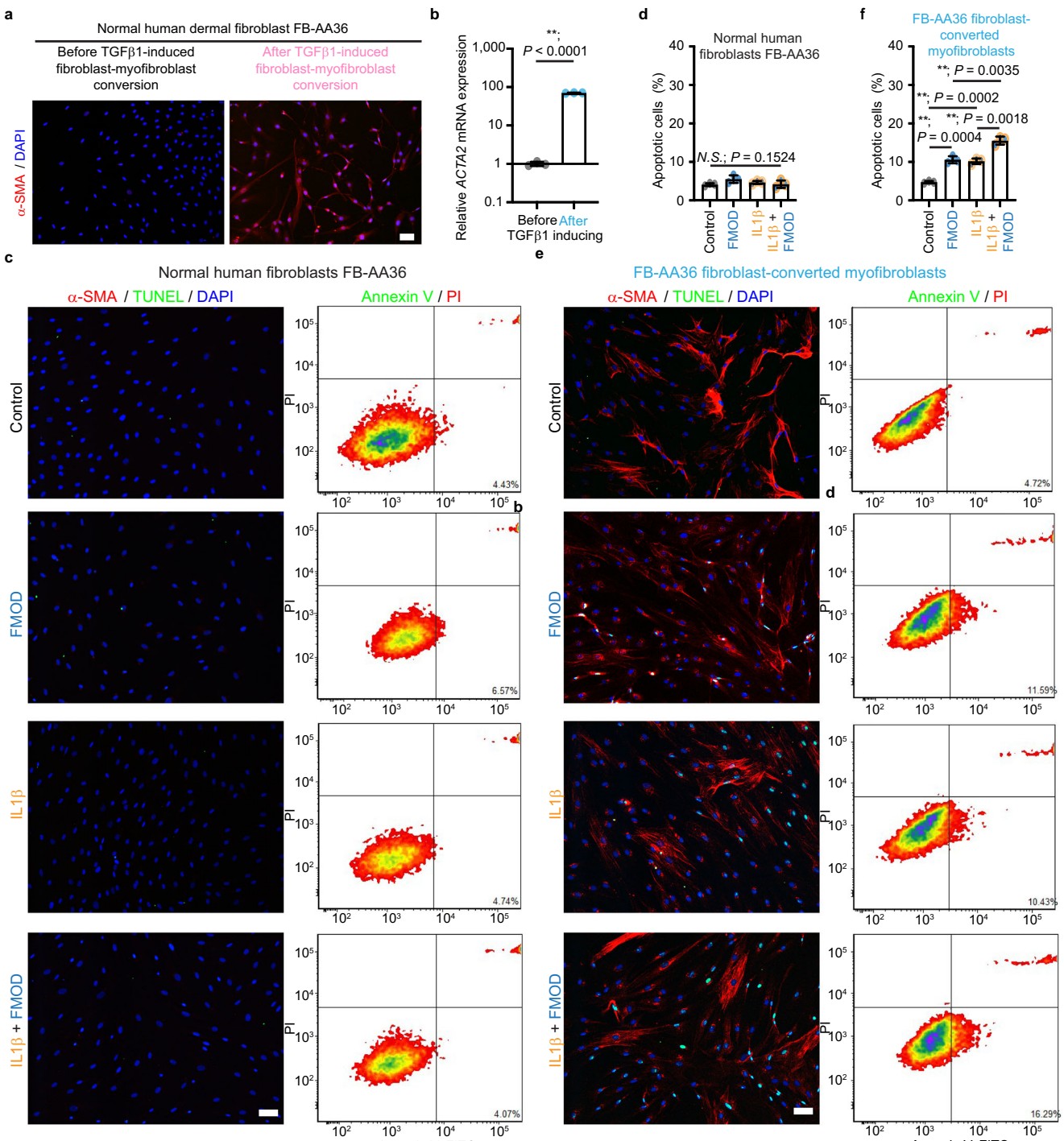

**Fig. 3 | Fibromodulin (FMOD) induces the apoptosis of the primary normal human dermal fibroblast (NHDF) FB-AA36-converted myofibroblasts.** All cells were subjected to serum starvation prior to treatment. **a** Representative images of the primary NHDF FB-AA36 (derived from a 36-year-old female African–American donor; Supplementary Table 1) stained with α-smooth muscle actin (α-SMA) by immunofluorescence staining before and after transforming growth factor (TGF) β1-induced fibroblast-myofibroblast conversion. **b** Expression of *ACTA2* in FB-AA36 fibroblasts before and after fibroblast-myofibroblast conversion. Data were normalized to the *ACTA2* level before TGFβ1-induced fibroblast-myofibroblast conversion. **c** Representative images of NHDF FB-AA36 with α-SMA and TUNEL staining, accompanied by the respective flow cytometry plots stained with Annexin V-FITC and PI staining. **d** Quantification of NHDF FB-AA36 apoptosis from flow cytometry in (**c**). **e** Representative images of FB-AA36 fibroblast-converted myofibroblasts with α-SMA and TUNEL staining, accompanied by the respective flow cytometry plots stained with Annexin V-FITC and PI staining. **f** Quantification of FB-AA36 fibroblast-converted myofibroblast apoptosis from flow cytometry in (**e**). Scale bars, 25 μm. Data presented as mean ± s.d. overlaying all the data points. *N* = 3 biological replicates; *P*-values were determined by two-tailed unpaired *t*-tests (**b**–**f**). *N.S.*, not significant, *P* > 0.05; **P* < 0.005. Source data are provided as a Source Data file.

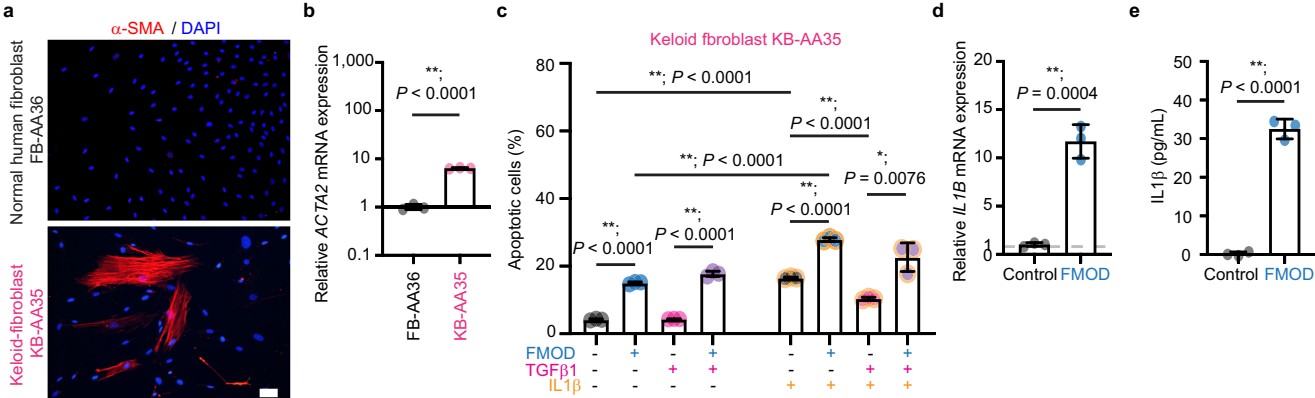

**Fig. 4 | Fibromodulin (FMOD) induces the apoptosis and interleukin (IL)1β expression of the human keloid-derived KB-AA35 fibroblasts.** All cells were subjected to serum starvation prior to treatment. **a** Representative images of the human keloid-derived fibroblast KB-AA35 (derived from a 35-year-old female African–American donor; Supplementary Table 1) stained with α-smooth muscle actin (α-SMA) by immunofluorescence staining. Scale bar, 25 μm. **b** Expression of *ACTA2* in FB-AA36 fibroblasts and KB-AA35 fibroblasts. Data were normalized to the *ACTA2* level of FB-AA36 fibroblasts. **c** Quantification of KB-AA35 fibroblast apoptosis by flow cytometry with Annexin V-FITC and PI staining. **d** Expression of *IL1B* in KB-AA35 fibroblasts with or without FMOD treatment. Data were normalized to the *IL1B* level without FMOD treatment. **e** Active IL1β production in KB-AA35 fibroblasts with or without FMOD treatment. Data presented as mean ± s.d. overlaying all the data points. $N = 3$ biological replicates; *P*-values were determined by two-tailed unpaired *t*-tests (**b**–**e**). *$P < 0.05$; **$P < 0.005$. Source data are provided as a Source Data file.

also bound to the membrane-associated IL1β-IL1R1-IL1RAP ternary complex (Fig. 8e). Surface plasmon resonance (SPR) provided evidence of moderately high binding affinities between FMOD and IL1R1 with a $K_D = 4.3 \times 10^{-7}$ M (Fig. 8f) and between FMOD and IL1RAP with a $K_D = 1.09 \times 10^{-6}$ M (Fig. 8g).

Sequence alignment has demonstrated that FMOD is highly conserved across mammalian species (Positives: 94.9%) and possesses a complex yet stable 3-dimensional structure[18] amenable to in silico analyses of protein-protein interactions. Aligned with the pull-down and SPR results, in silico modeling indicates that FMOD binds to the IL1β-IL1R1 binary complex (Fig. 8h) and IL1β-IL1R1-ILRAP ternary complex (Fig. 8i) on the convex surface of FMOD's core protein near the C-terminal region. Notably, the predicted binding sites for the IL1β-IL1R1 and IL1β-IL1R1-ILRAP complexes do not overlap with known FMOD binding sites for collagen, lysyl oxidase, complement element C1q, and TGFβs[18,40–45]. Importantly, the predicted FMOD binding did not inhibit the formation of IL1β binary and ternary complexes. Instead, it significantly increased the total number of hydrophobic interactions between lL1β and its receptor and co-receptor (Supplementary Fig. 16). For instance, while FMOD binding slightly decreased the number of hydrogen bonds between IL1β and IL1R1, it facilitated the formation of more hydrogen bonds between IL1β and IL1RAP. Therefore, FMOD markedly reduced the binding energy among the components of the complex (Fig. 8j) and shortened the average distance of responsive amino acids between lL1β and IL1R1, as well as between lL1β and ILRAP (Fig. 8k), thereby facilitating the easier and tighter complex formation.

Given that IL1β-IL1R1 binary complex formation is the initial step in IL1β signaling[36], we explored how FMOD might influence this process. A proximity ligation assay (PLA) was used to visualize the direct binding between IL1R1 and IL1β on the surface of BJ-myofibroblasts. Both exogenous IL1β and FMOD stimulated IL1R1:IL1β binding on the BJ-myofibroblast surface as early as 1 h post-treatment, peaking at 48 h (Fig. 9). During the initial 48 h post-treatment period, exogenous IL1β and FMOD alone resulted in comparable IL1R1:IL1β binding levels on the BJ-myofibroblast surface, as indicated by the number of PLA signals per cell and the percentage of PLA$^{HIGH}$ cells. However, the combined application of IL1β and FMOD synergistically enhanced IL1R1:IL1β binding. Notably, at 72 h post-treatment, more IL1R1:IL1β binding

remained in the FMOD group than in the exogenous IL1β group (Fig. 9g).

Upon formation, the IL1β-IL1R1 binary complex recruits IL1RAP, forming a ternary complex to initialize IL1β signaling transduction[36]. IL1RAP binds to the binary complex via its backside through extensive contact with IL1R1[36]. Notably, the overall architecture of the binary complex remains predominately unchanged during ternary complex formation[36]. To determine the effect of FMOD on the recruitment of IL1RAP, we used PLA of IL1R1:IL1RAP binding, representing the formation of the IL1β-IL1R1-ILRAP ternary complex (Fig. 10). FMOD significantly induced IL1R1:IL1RAP binding on the BJ-myofibroblast surface within 1 h, while exogenous IL1β only induced negligible binding. Although both FMOD- and IL1β-induced IL1R1:IL1RAP binding peaked at 24 h, FMOD-induced IL1R1:IL1RAP-binding maintained the same level at 48 h, whereas that induced by IL1β decreased. Furthermore, the FMOD-treated group displayed more binding signals per cell and a higher percentage of IL1R1:IL1RAP-PLA$^{HIGH}$ BJ-myofibroblasts than the IL1β-treated group at the peak time point, 24 h post-treatment (Fig. 10h). Taken together, FMOD accelerated and prolonged the formation of the IL1β-IL1R1-ILRAP ternary complex on the surface of myofibroblasts, which in turn, enhanced IL1β autoinduction and led to increased myofibroblast apoptosis.

## Discussion

Myofibroblasts play a crucial role in restoring homeostasis and tissue integrity following traumatic injury and thus are often referred to as *'professional'* repair cells[46]. Despite their phenotypic and functional heterogeneity, common characteristics of dermal injury-activated myofibroblasts include developing contractile force transmission to the ECM and promoting wound defect contraction and closure[1,10]. Once the cutaneous wound is re-epithelialized and *'closed,'* activated myofibroblasts are removed via deactivation, senescence, and apoptosis, with apoptosis being the typical and major route for myofibroblast elimination[1,2,10]. The specified contractile ability of myofibroblasts induces more strain and compaction of the ECM, increasing microenvironmental force that can not only further promote myofibroblast activation but also inhibit myofibroblast apoptosis[7,10]. Consequently, the persistence of myofibroblasts, rather than their activation, is considered ultimately responsible for pathological scar formation and many pathological fibrotic conditions[7,10]. For example, myofibroblast

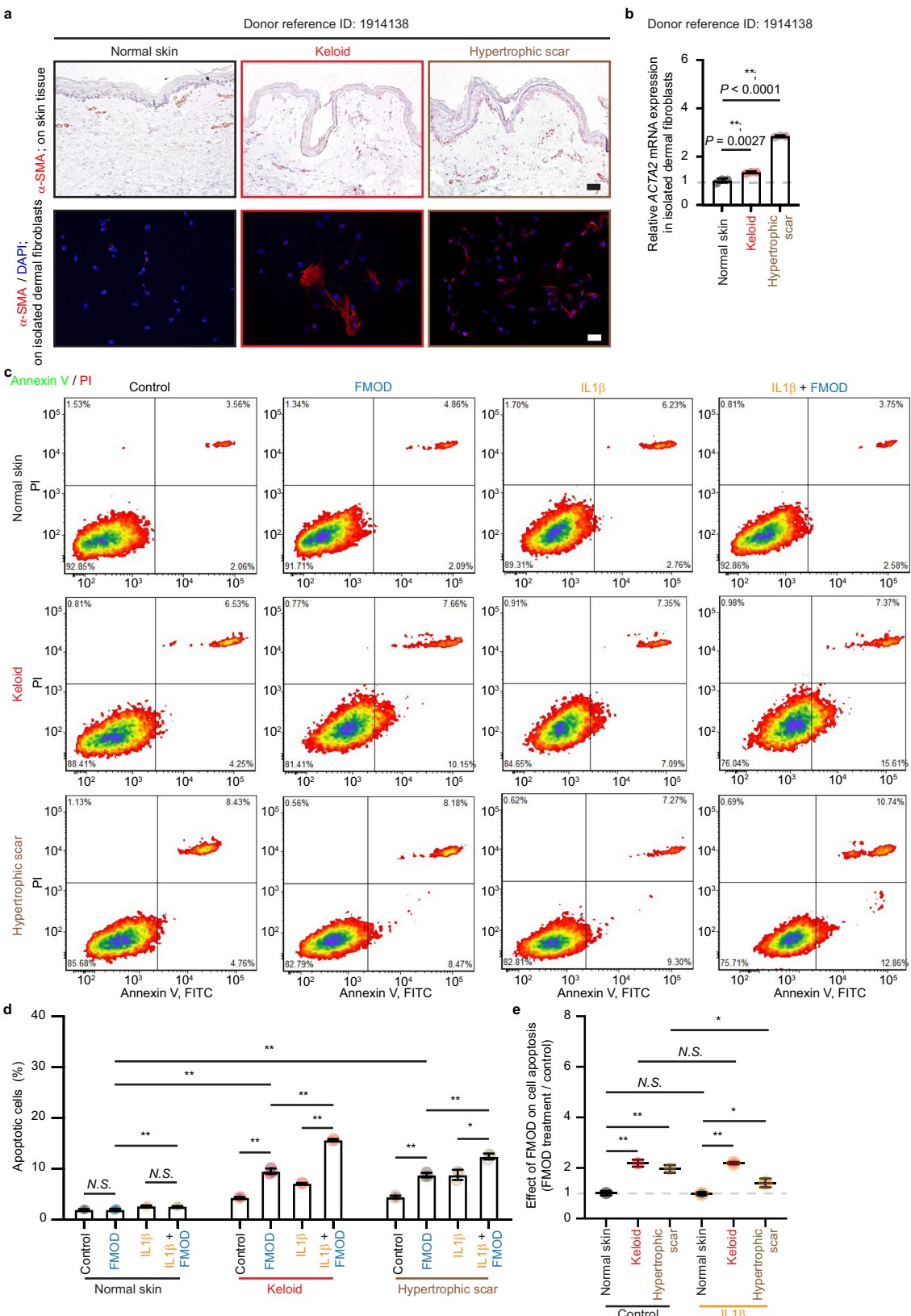

persistence can even cause scarless fetal wounds to heal with scar[3]. Meanwhile, pro-survival growth factors (such as TGFβ1) and bio-mechanical factors (such as ECM stiffness and mechanical loading) can decrease myofibroblast apoptosis, leading to increased tissue fibrosis[2,7,10]. On the other hand, expedited myofibroblast clearance shortens dysfunctional ECM deposition, potentially triggering earlier

and more optimal ECM remodeling, thus reducing scar formation while enhancing wound healing outcomes[13]. Consequently, inducing myofibroblast apoptosis is a broadly recognized strategy today to prevent or treat tissue fibrosis[2,7,10].

Apoptosis can be triggered by the two interconnected pathways, the extrinsic and intrinsic pathways[10]. Previous studies have revealed

**Fig. 5 | Fibromodulin (FMOD) induces apoptosis in dermal fibroblasts derived from the keloid and hypertrophic scar but not in the normal skin tissue of donor 1914138.** All cells were subjected to serum starvation prior to treatment. **a** Representative images of normal skin (from abdomen), keloid (from abdomen), and hypertrophic scar (from left shoulder) tissues collected from the donor with the National Disease Research Interchange (NDRI) reference ID 1914138 with α-smooth muscle actin (α-SMA) staining by immunohistochemistry (IHC; upper), and the representative images of dermal fibroblasts isolated from these tissues with α-SMA staining by immunofluorescence staining (lower). **b** Expression of *ACTA2* in dermal fibroblasts derived from tissues described from (**a**). Data were normalized

to the *ACTA2* level of fibroblasts isolated from the normal skin. **c** Representative flow cytometry plots of fibroblasts derived from tissues described in (**a**) with Annexin V-FITC and PI staining. **d** Quantification of apoptotic dermal fibroblasts by flow cytometry from (**c**). **e** The influence of FMOD on cell apoptosis with and without IL1β was calculated from (**d**). Scale bars, 50 μm (black) or 25 μm (white). Data presented as mean ± s.d. overlaying all the data points. *N* = 3 biological replicates; *P*-values were determined by two-tailed unpaired *t*-tests (**b**–**e**). *N.S.*, not significant, *P* > 0.05; *\*P* < 0.05; *\*\*P* < 0.005. Source data are provided as a Source Data file.

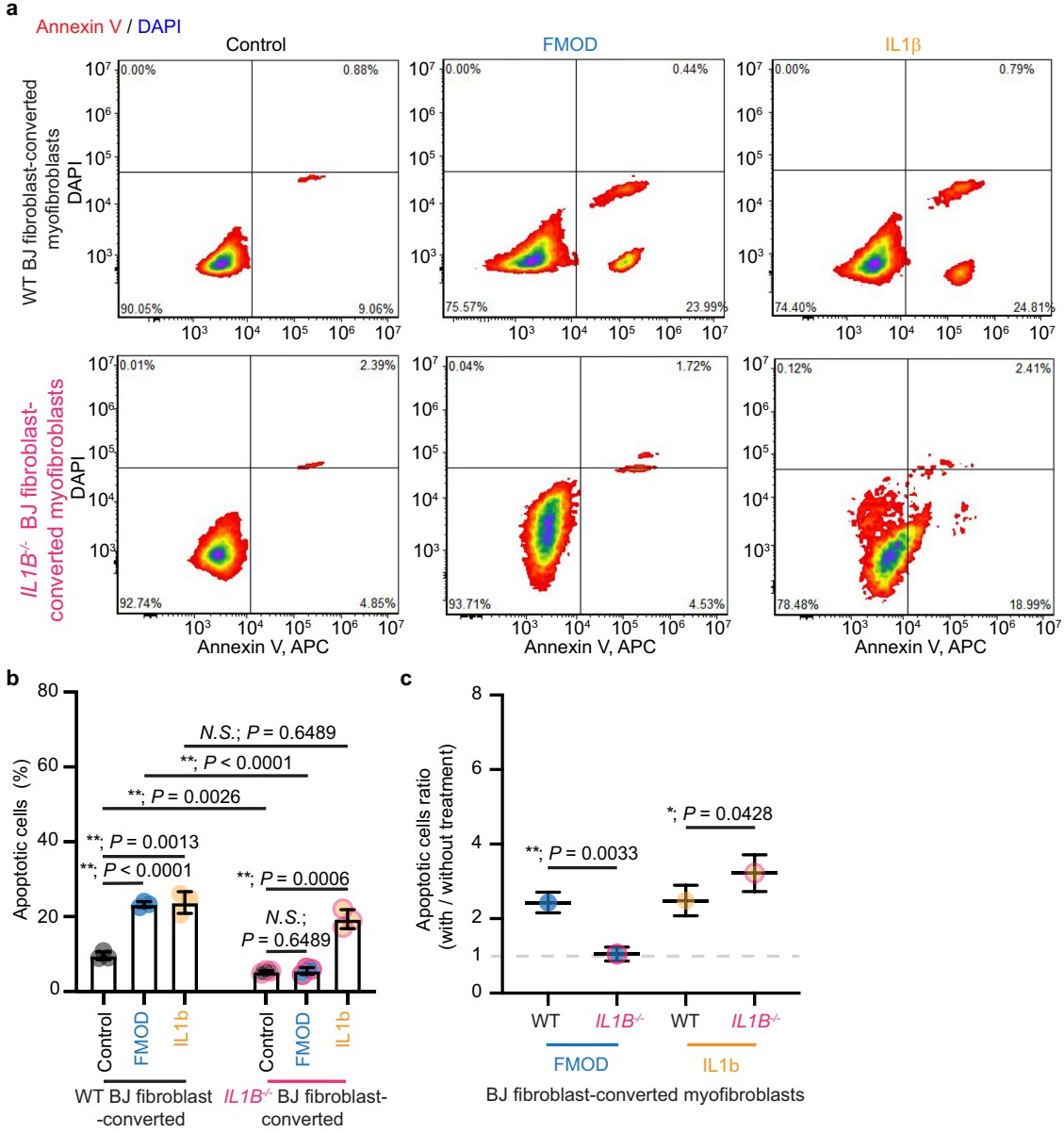

**Fig. 6 | *Interleukin (IL)1B-knockout* diminishes the pro-apoptosis effect of FMOD on BJ-myofibroblasts.** All cells were subjected to serum starvation prior to treatment. **a** Representative plots of wildtype (WT) and *IL1B⁻/⁻* BJ-myofibroblasts stained with Annexin V-allophycocyanin (APC) and DAPI staining. **b** Quantification of BJ-myofibroblast apoptosis from (**a**). **c** Impacts of fibromodulin (FMOD) and IL1β on

cell apoptosis determined from (**b**). Data presented as mean ± s.d. overlaying all the data points. *N* = 3 biological replicates; *P*-values were determined by two-tailed unpaired *t*-tests (**b**, **c**). *N.S.*, not significant, *P* > 0.05; *\*\*P* < 0.005. Source data are provided as a Source Data file.

that the extrinsic pathway is suppressed in myofibroblasts[10]. Conversely, fibroblast-to-myofibroblast conversion increases the mitochondrial priming of these cells[6,10,47], indicating myofibroblasts are primarily predisposed to apoptosis (i.e., apoptosis-primed) via the

intrinsic pathway (i.e., mitochondrial apoptosis)[10]. Notably, while fibroblasts are deemed 'unprimed' for apoptosis and exhibit resistance to pro-apoptotic factors such as IL1β, this cytokine can instigate targeted apoptosis of myofibroblasts through various mechanisms[10,25].

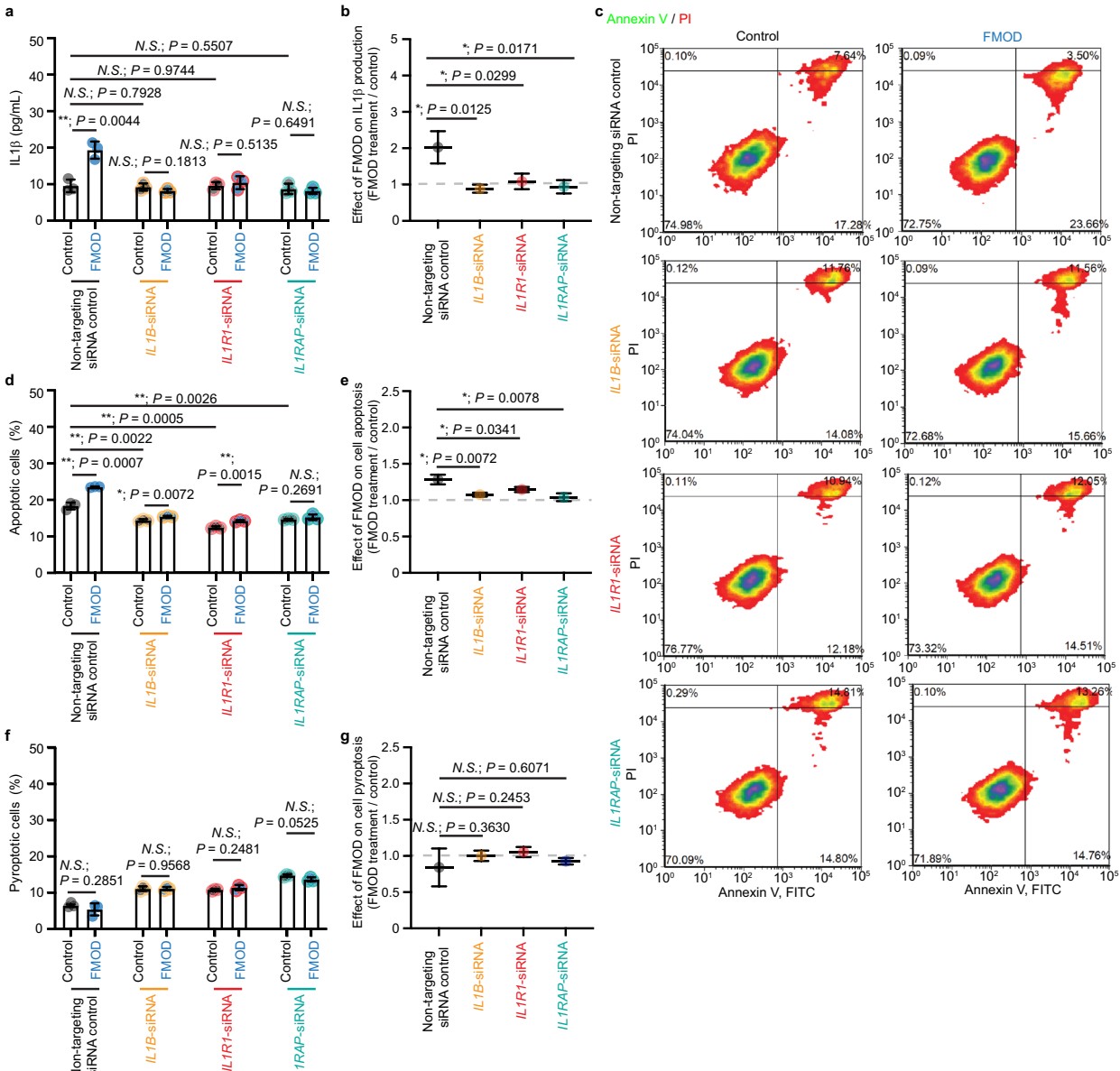

**Fig. 7 | Knockdown of *interleukin (IL)1B, interleukin 1 receptor type 1 (IL1R1)*, or *interleukin-1 receptor accessory protein (IL1RAP)* expression weakens the effects of fibromodulin (FMOD) on myofibroblast IL1β expression and apoptosis.** All cells were subjected to serum starvation prior to treatment. **a** ELISA assessment of active IL1β production from the *IL1B*-, *IL1R1*-, and *IL1RAP*-knockdown BJ-myofibroblasts, as well as the control cells transfected with non-targeting siRNA control, respectively. **b** Impacts of FMOD on active IL1β production determined from (**a**). **c** Representative flow cytometry plots of BJ-myofibroblasts described in (**a**). Cell apoptosis was evaluated with staining of Annexin V-FITC and PI staining. **d** Quantification of apoptotic BJ-myofibroblasts by flow cytometry from (**c**). **e** Impacts of FMOD on BJ-myofibroblast apoptosis determined from (**d**). **f** Quantification of pyroptotic BJ-myofibroblasts by flow cytometry from (**c**). **g** Impacts of FMOD on BJ-myofibroblast pyroptosis determined from (**f**). Data presented as mean ± s.d. overlaying all the data points. $N = 3$ biological replicates; $P$-values were determined by two-tailed unpaired $t$-tests (**a**–**h**). *N.S.*, not significant, $P > 0.05$; *$P < 0.05$; **$P < 0.005$. Source data are provided as a Source Data file.

The current study demonstrates that exogenous FMOD in adult dermal wounds accelerates clearance of α-SMA⁺ myofibroblast after wound closure in comprehensive mouse, rat, and pig models, the latter being a gold standard for human wound healing. Mechanistically, FMOD can significantly lower the binding energy for IL1β binding with its cognate receptor IL1R1 and co-receptor IL1RAP, as revealed by in silico analyses. This finding indicates that even at low levels of endogenous IL1β expression, FMOD facilitates sufficient formation of IL1β-IL1R1 binary complex and IL1β-IL1R1-IL1RAP ternary complex, boosting the IL1β autocrine loop and other IL1β downstream signal transductions[48,49]. The resulting increase in IL1β autoinduction disrupts the balance between downstream pro- and anti-apoptotic proteins, ultimately leading to the targeted myofibroblast apoptosis when mitochondrial priming is sufficiently high to surpass the apoptotic threshold (Supplementary Fig. 17).

Meanwhile, TGFβ1 inhibits mitochondrial apoptosis in myofibroblasts and counteracts the pro-apoptosis effects of IL1β[10]. Notably, through its N-terminus, FMOD can directly bind to TGFβ1 and its latent form[18,40], potentially sequestering TGFβ1 globally and preventing it from engaging with its receptors and activating downstream signal transduction[40] that allows myofibroblasts to escape apoptosis. Our previous research highlights FMOD's ability to selectively dampen TGFβ1's non-Smad-driven pathways[13], which are predominantly responsible for TGFβ1's anti-apoptosis activities[10]. Remarkably, the

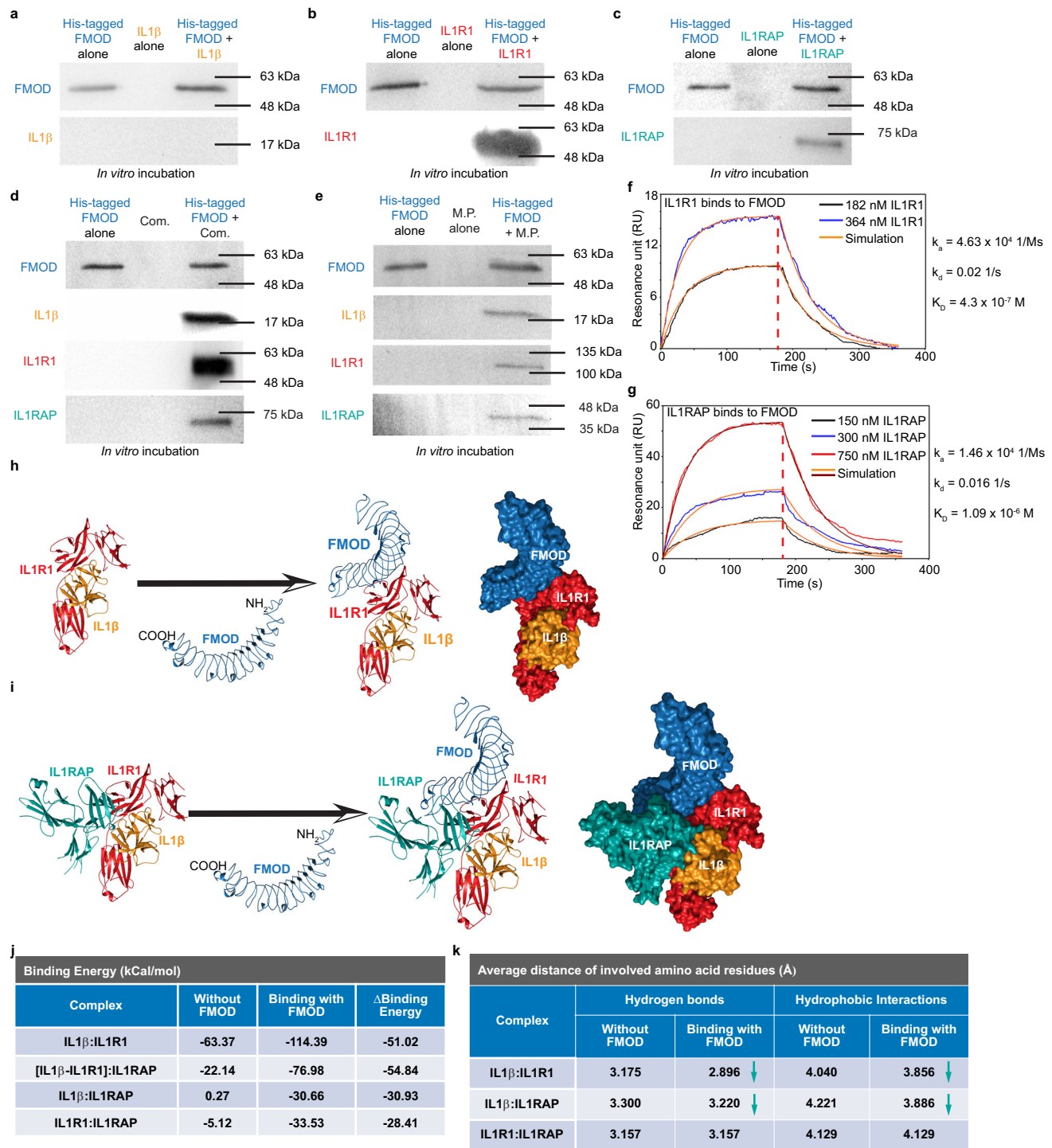

**Fig. 8 | Fibromodulin (FMOD) binds to the interleukin (IL)1β-interleukin 1 receptor type 1 (IL1R1)-interleukin-1 receptor accessory protein (IL1RAP) ternary complex. a** Image depicting the pull-down assay conducted with recombinant human His-tagged FMOD, which was incubated in vitro with recombinant IL1β. **b** Image depicting the pull-down assay conducted with recombinant human His-tagged FMOD, which was incubated in vitro with recombinant IL1R1. **c** Image depicting the pull-down assay conducted with recombinant human His-tagged FMOD, which was incubated in vitro with recombinant IL1RAP. **d** Image depicting the pull-down assay conducted with recombinant human His-tagged FMOD, which was incubated in vitro with IL1β-IL1R1-IL1RAP ternary complex (Com.) in an acellular system. **e** Image depicting the pull-down assay conducted with recombinant human His-tagged FMOD, which was incubated in vitro with the whole membrane protein

(M.P.) extracted from BJ-myofibroblasts. Blot is representative of $N = 2$ biological replicates (**a–e**). **f** Surface plasmon resonance (SPR) spectrum characterizing the binding properties between IL1R1 and FMOD. **g** SPR spectrum characterizing the binding properties between IL1RAP and FMOD. **h** In silico analysis of protein-protein interactions between FMOD with IL1β-IL1R1 binary complex. **i**. In silico analysis of protein-protein interactions between FMOD and IL1β-IL1R1-IL1RAP ternary complex. **j** In silico analysis of binding energy among IL1β-IL1R1-IL1RAP ternary complex components. **k** In silico analysis of the average distance of the involved amino acid residues of IL1β-IL1R1-IL1RAP ternary complex. More details about the amino acid residue interactions are demonstrated in Supplementary Fig. 16. Source data are provided as a Source Data file.

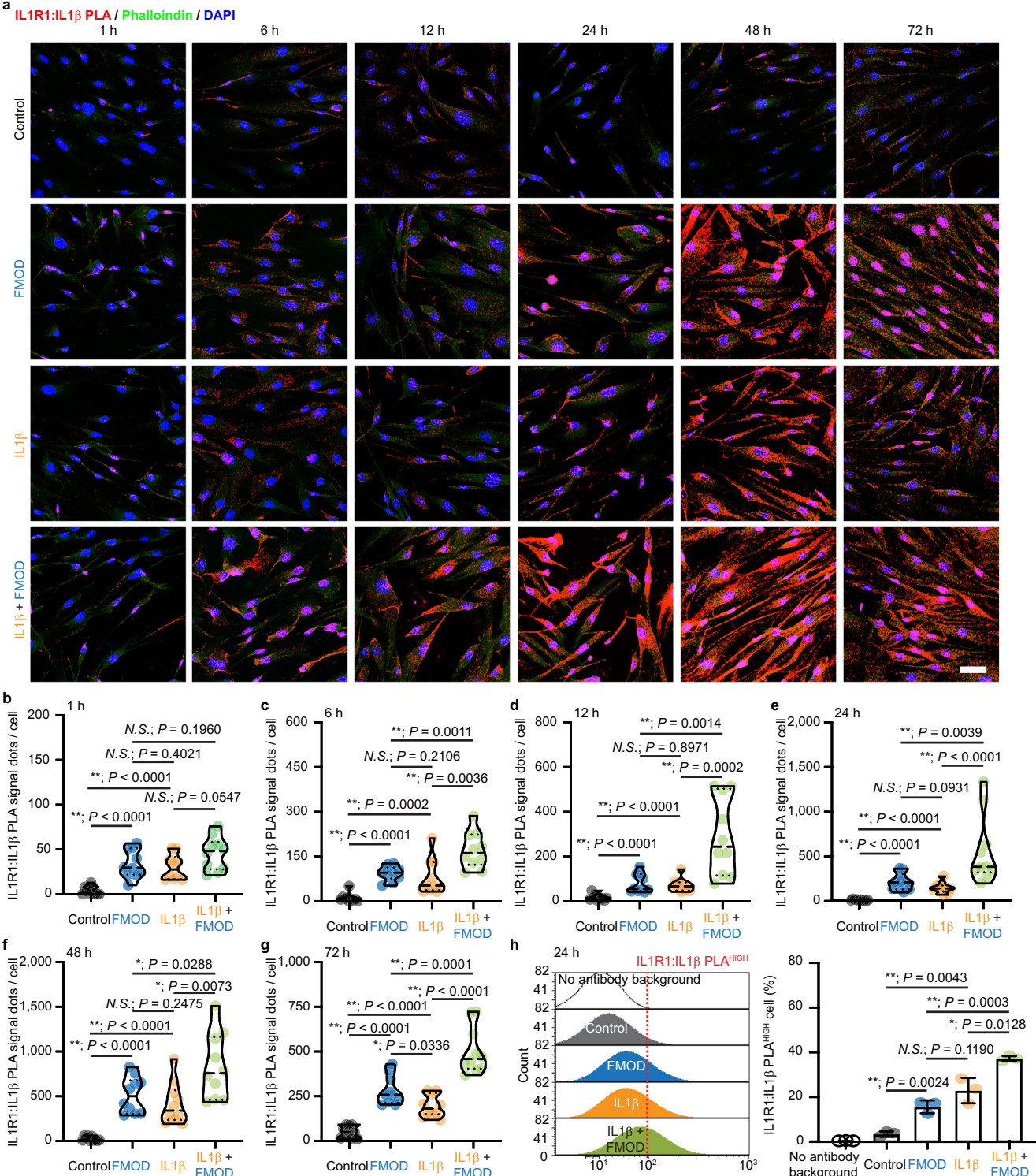

**Fig. 9 | Fibromodulin (FMOD) promotes the direct binding of interleukin 1 receptor type 1 (IL1R1) and interleukin (IL)1β on the surface of BJ-myofibroblasts.** All cells were subjected to serum starvation prior to treatment. **a** Representative images of in situ IL1R1:IL1β-proximity ligation assay (PLA) staining on attached BJ-myofibroblasts. Phalloindin was used to stain F-actin, while DAPI was used for nuclei staining. **b**–**g** Quantification of IL1R1:IL1β-PLA⁺ signal per cell at 1- (**b**), 6- (**c**), 12- (**d**), 24- (**e**), 48- (**f**), and 72-h (**g**) from (**a**). **h** Representative flow cytometry plot and quantification of IL1R1:IL1β-PLA staining of BJ-myofibroblasts at 24 h post-treatment. Scale bar, 50 µm. Data presented as violin plots (**b-g**) or mean ± s.d. **h** overlaying all the data points. N = 10 (**a–g**) or 3 (**h**), respectively; P-values were determined by two-tailed Mann–Whitney U tests (**b–g**) or two-tailed unpaired t-tests (**h**), respectively. N.S., not significant, P > 0.05; *P < 0.05; **P < 0.005. Source data are provided as a Source Data file.

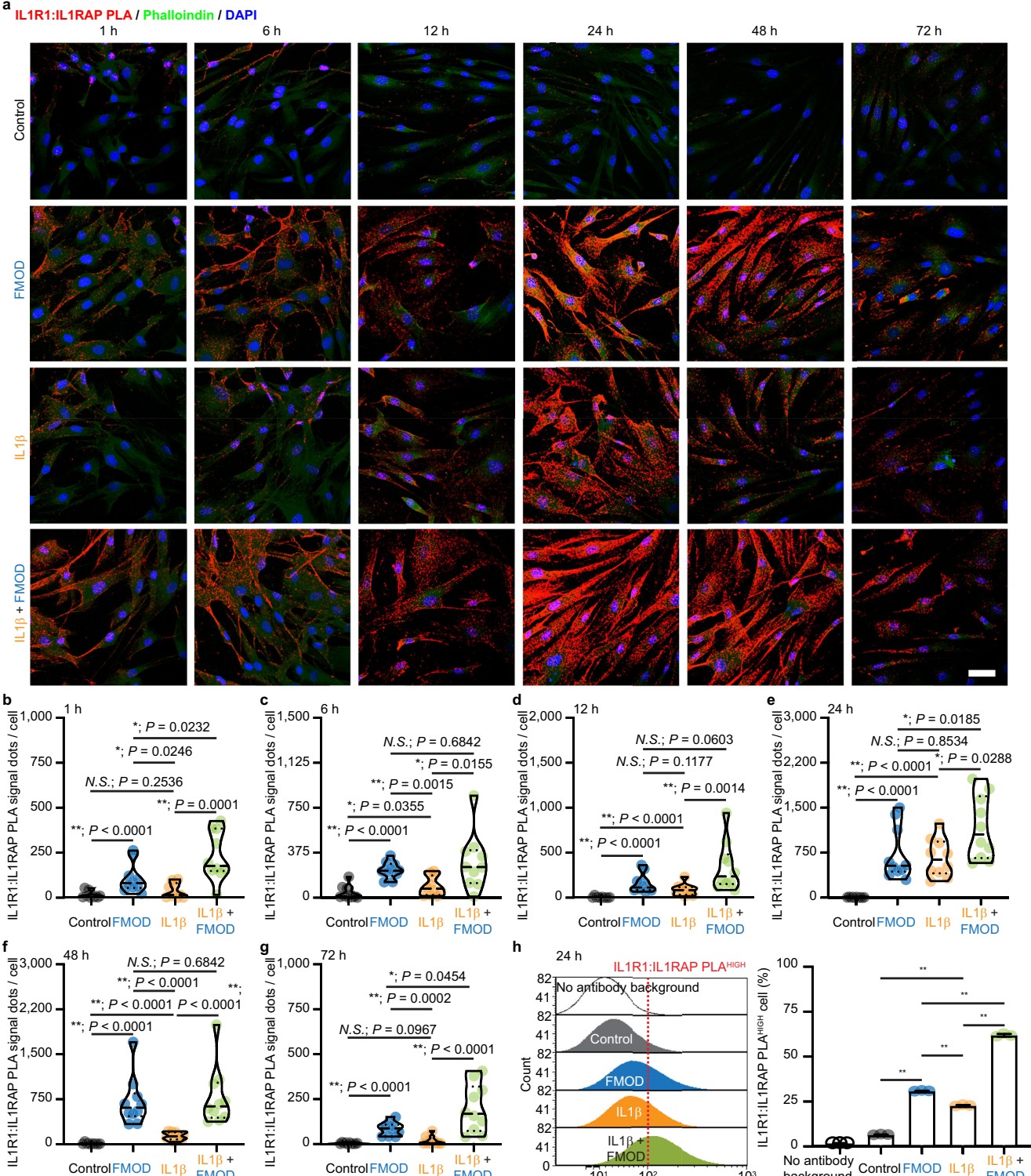

**Fig. 10 | Fibromodulin (FMOD) promotes the direct binding of interleukin 1 receptor type 1 (IL1R1) and interleukin-1 receptor accessory protein (IL1RAP) on the surface of BJ-myofibroblasts.** All cells were subjected to serum starvation prior to treatment. **a** Representative images of in situ IL1R1:IL1RAP-proximity ligation assay (PLA) staining on attached BJ-myofibroblasts. Phalloindin was used to stain F-actin, while DAPI was used for nuclei staining. **b**–**g** Quantification of IL1R1:IL1RAP-PLA[+] signal per cell at 1- (**b**), 6- (**c**), 12- (**d**), 24- (**e**), 48- (**f**), and 72-h (**g**) after the treatment of FMOD and/or IL1β from (**a**). **h** Representative flow cytometry plot and quantification of IL1R1:IL1RAP-PLA staining of BJ-myofibroblasts at 24 h post-treatment. Scale bar, 50 μm. Data presented as violin plots (**b**–**g**) or mean ± s.d. (**h**) overlaying all the data points. N = 10 (**a**–**g**) or 3 (**h**), respectively; P-values were determined by two-tailed Mann-Whitney U tests (**b**–**g**) or two-tailed unpaired t-tests (**h**). N.S., not significant, P > 0.05; *P < 0.05; **P < 0.005. Source data are provided as a Source Data file.

TGFβ1-binding sites on FMOD do not overlap with the sections of FMOD that boost and stabilize the formulation of IL1β-IL1R1-IL1RAP ternary complex, indicating that FMOD might simultaneously enhance IL1β signaling while weakening TGFβ1 non-Smad-driven signaling, thereby exerting a dual effect that encourages myofibroblast apoptosis (Supplementary Fig. 17). The delicate modulation of TGFβ1 signaling by FMOD explains why TGFβ1 fails to mitigate the FMOD-induced autoinduction of IL1β and targeted apoptosis of myofibroblasts. Additionally, it clarifies how FMOD counteracts TGFβ1's effects to prevent myofibroblast apoptosis. Moving forward, it is vital to conduct future investigations to elucidate FMOD's influence on the intricate crosstalk among non-Smad-driven signaling transduction downstream of TGFβ1[10,50].

In alignment with our previous study that revealed the ability of FMOD to significantly reduce skin scarring in adults by eliciting fetal-like behaviors in adult dermal fibroblasts and wounds, including a swift activation and conversion of fibroblasts into myofibroblasts and an increase in cellular mobility and contractility[13,18,51,52], the current study demonstrated FMOD's capability for accelerating the apoptosis and clearance of myofibroblasts, another characteristic observed in scarless fetal cutaneous wounds[3,5,12]. These two cellularly distinct effects of FMOD on myofibroblast formation and elimination highlight the importance of timing and the inherent complexity within biological systems. However, the finding that FMOD effectively induces myofibroblast apoptosis potentially paves the pathway for developing alternative, more efficient existing scar treatments.

In addition, our current data offers compelling evidence to suggest that, albeit not activated yet, human keloid- and HS-derived fibroblasts, or at least a subpopulation of these cells, can replicate the targeted apoptosis of BJ- and NHDF-converted myofibroblasts in response to IL1β and FMOD. This finding indicates that a significant proportion, if not all, of keloid- and HS-derived fibroblasts are proto-myofibroblasts or prone-to-be myofibroblasts, whose apoptosis can be selectively induced by FMOD. Undoubtedly, a more comprehensive comparison among dermal fibroblasts derived from normal skin, keloid, and HS tissues from donors of broadly diverse races, genders, and ages, as well as different tissue locations, will further aid our understanding of FMOD's anti-scarring potency. It is important to note that none of the currently available animal models can fully replicate the pathology of human keloid and HS formation[29,53,54]. While we eagerly await a suitable animal model to confirm if FMOD-induced targeted myofibroblast apoptosis could be adopted to therapeutically treat established scars, the current in vitro data bolsters our confidence in this novel strategy.

Besides promoting HS and keloids from the cutaneous organ system, myofibroblast evasion from apoptosis also causes fibrotic diseases in other organ systems. For instance, numerous studies have highlighted that activated myofibroblasts are the primary contributor to pulmonary fibrosis, which can be triggered by injuries or infections such as severe acute respiratory syndrome (SARS), Middle East respiratory syndrome (MERS) and coronavirus disease 2019 (COVID-19)[10,55,56]. Given the high potential for lung fibrotic diseases to progress[55], eliminating persistent myofibroblasts is crucial for promoting resolution[57]. Notably, the potency of IL1β in promoting myofibroblast apoptosis was first recognized in lung cells[58], indicating that FMOD may also have significant potential to help prevent or even reverse possibly pulmonary fibrosis. Moreover, FMOD-induced myofibroblast apoptosis could also be beneficial for other pathologies associated with myofibroblast persistence, including autoimmune rheumatic diseases, such as systemic sclerosis, rheumatoid arthritis, as well as liver, kidney, and muscle fibrosis[10,59].

Meanwhile, it is also important to acknowledge the current study's limitations. First, while animal excision wound models offer valuable insights into the fundamental processes of wound healing and scar formation, they are simplified representations that may not encompass all the complexities of human wound healing and scar formation. For instance, deep partial-thickness burn injuries, which commonly result in HS formation, exhibit more complex inflammatory responses and healing dynamics that could significantly influence the healing outcome and the extent of scarring[60,61]. Given that deep partial-thickness burns can also induce fibroblasts to convert into myofibroblasts, contributing to fibrotic scarring[62], it remains to be validated whether FMOD's ability to selectively expedite myofibroblast clearance from the wound area extends to deep partial-thickness burn injuries. Second, this study focused on dermal fibroblasts. Besides local dermal fibroblasts, myofibroblasts are known to originate from other precursors, such as local mesenchymal stem cells, bone marrow-derived mesenchymal stem cells, and cells derived from an epithelial-mesenchymal transition process[46,63]. Therefore, whether FMOD promotes apoptosis of myofibroblasts derived from these cells should also be confirmed. Third, investigations have shed light on how the equilibrium among the B cell leukemia/lymphoma 2 (BCL2) family members regulates mitochondrial apoptosis[6,10,47,64]. This family can be categorized into four subfamilies: _effectors_ that initiate apoptosis by inducing mitochondrial outer membrane permeabilization (MOMP), _pro-apoptotic activators_ that bind and activate effectors, _pro-survival/anti-apoptotic proteins_ that bind to and block effectors and activators, and _sensitizers_ that bind to pro-survival proteins, thereby freeing activators[6,10,47,64]. In an apoptosis-primed state, such as in myofibroblasts, the downregulation of crucial pro-survival proteins can trigger swift apoptosis[6,10,47]. For example, IL1β lower the levels of pro-survival protein BCL2 while interfering with focal adhesion kinase (FAK) signaling, which inhibits pro-apoptotic sensitizer BCL2 associated agonist of cell death (BAD), consequently promoting myofibroblast apoptosis (Supplementary Fig. 17). Conversely, TGFβ1, via its non-Smad-driven pathways, augments the expression of pro-survival BCL2 proteins while suppressing the pro-apoptotic sensitizer BAD (Supplementary Fig. 17), representing one mechanism by which TGFβ1 aids myofibroblasts in escaping apoptosis[10]. Comprehending how FMOD unbalances BCL2 family members to effectively induce myofibroblast apoptosis, especially in the context of IL1β and/or TGFβ1, is an important area for future studies. It may provide valuable insights on how to harness FMOD's full potential in modulating myofibroblast apoptosis to develop better strategies for improving wound healing and reducing fibrotic tissue formation. Fourth, recent research has shown that several caspases associated with apoptosis, including caspase-3 and -8, can trigger pyroptosis by cleaving gasdermin E, thereby causing an apoptosis-to-pyroptosis shift[27,28]. Consequently, even though FMOD does not initially induce pyroptosis, it is conceivable that apoptosis induced by FMOD could subsequently lead to a secondary pyroptosis in vivo, which could also contribute to the expedited clearance of myofibroblasts from the wound site, warranting further investigations. Furthermore, the concept of 'myofibroblast reversal' adds another layer of complexity to our understanding of tissue repair. This process involves the transformation of activated myofibroblasts into scar-resolving cells and necessitates a transition stage whereby the cyclin-dependent kinase inhibitor 2 A (CDKN2A; _aka_., p16[INK4a]) is expressed[10]. Interestingly, FMOD has been shown to upregulate CDKN2A in dermal fibroblasts[15,65], which raises an intriguing question for future research: Does FMOD also mediate myofibroblast reversal, thereby offering additional benefits against fibrotic conditions?

In conclusion, as a natural ECM component broadly distributed in the connective tissues, FMOD appears to be a promising promoter of myofibroblast apoptosis that could potentially be more effective and safer at reducing organ fibrosis. We hope the current study will provide the impetus to develop and use FMOD-based strategies for myofibroblast clearance strategies in the fight against fibrosis. These findings will also serve as a valuable reference for deepening the understanding of FMOD and other SLRPs in the context of fibrotic disease prevention, progression, and treatment.

## Methods

### Ethical statement

All animal surgeries in this study were performed following the United States National Institute of Health (NIH) Guide for the Care and Use of Laboratory Animals under protocols thoroughly reviewed and approved by the Chancellor's Animal Research Committee at the University of California, Los Angeles (UCLA; protocol number: 2000-058 and 2008-016), ensuring compliance with ethical standards for animal research. Human skin samples were procured by NDRI (Philadelphia, PA; the researcher code is DTIK2). According to the NDRI's policy, the signed consents/authorizations were kept on file at the tissue acquisition site without being disclosed to the investigators. Since the donated tissues are medically discarded wastes and no personally identifiable information is provided, UCLA institutional review boards (IRB) determined their usage was an exception for IRB review (Application number: IRB#19-000605).

### FMOD production

cDNA of a human FMOD transcript (Genbank assessor number: NM_002023) was subcloned into a commercially available vector pSecTag2A (Life Technology, Grand Island, NY) with C-terminal His-tag and transfected into CHO-K1 cells (ATCC, Manassas, VA)[15]. After establishing a stable expression clone, the FMOD was produced and purified by a contract research organization, GenScript (Piscataway, NJ). Briefly, a stable human recombinant FMOD-expressing CHO-K1 cell line was cultured in a 1 L serum-free Freestyle CHO Expression Medium (Thermo Fisher Scientific, Canoga Park, CA) at 37 °C, 5% $CO_2$ in an Erlenmeyer flask. Cell culture supernatant was harvested on day 10 for purification with HiTrap™ IMAC HP, 1 mL column (GE Healthcare, Uppsala, Sweden). The fractions from a 100 mM imidazole elution were collected and dialyzed against 20 mM phosphate-buffered saline (PBS), pH 7.4. After that, the sample with low conductivity was loaded onto HiTrap™Q HP 1 mL column (GE Healthcare) for further purification. FMOD was then purified under non-reducing conditions, dialyzed again[16], and then subjected to lyophilization. FMOD was reconstituted in PBS and then sterilized through a 0.22 μm filter (Thermo Fisher Scientific) before usage[65].

### Adult mouse skin wound model

All rodents were housed under a 12 hour light/12 hour dark cycle, with a controlled temperature range of 22–26 °C and a relative humidity level of 40–60%. 3 month old male 129/sv WT and *Fmod*[-/-] (B6;129-FMOD<tm1Aol >/SooJ, which is now available at the Jackson Laboratory Repository with the JAX Stock No. 037010) mice were anesthetized, and the dorsal skin was sterilely prepared. Four full-thickness, 10 mm x 3 mm skin ellipses, with the underlying *panniculus carnosus* muscles, were excised on each mouse. Each open wound edge was injected with 25 μL PBS (control) or 0.4 mg/μL FMOD in PBS (25 μL x 2 edges = 50 μL total/wound). Wounds were then primarily closed with 5-0 Nylon using two simple interrupted sutures consistently placed at one-third intervals in each 10 mm length wound. To minimize adjacent wound effects, all wounds were separated by at least 2 cm. Sutures were removed 7 days post-injury, and wounds were harvested 14 days post-injury[13].

### Adult rat skin wound model

Adult male Sprague-Dawley rats (weighing ~300 g) were anesthetized, and the dorsal skin was sterilely prepared. Six full-thickness, 10 mm x 3 mm skin ellipses, with the underlying *panniculus carnosus* muscles, were excised on the dorsum of each animal. Each open wound edge was injected with 25 μL PBS (control) or 25 μL 0.4 mg/μL FMOD in PBS (25 μL x 2 edges = 50 μL total/wound). Wounds were then marked with permanent dye and closed primarily with 4–0 Nylon using two simple interrupted sutures consistently placed at one-third intervals in each 10 mm length wound. All wounds were separated by at least 2 cm to

minimize adjacent wound effects. Sutures were removed 1 week after injury. Wounds were harvested 2 weeks after injury and divided in two along their short axes[13].

### Excessive-mechanical-loading porcine wound model

20 kg female red Duroc pigs (Pork Power Farms, Turlock, CA) were sedated with Telazol® (Tiletamine and Zolazepam; Fort Dodge Animal Health, Fort Dodge, IA), a 5 mg/kg intramuscular injection. The pigs were then endotracheally intubated and maintained under a surgical plane of anesthesia with isoflurane at 0.5-2.5% in room air. The flank and back hair was clipped, and the skin was sterilely prepared with three alternating scrubs of povidone-iodine solution and alcohol. Since porcine skin is significantly thinker than rodent skin, full-thickness wounds, down to the fascia, were created with a #15 surgical blade by excising a 1.5 cm length x 2.0 cm width ellipse of skin with its long axis running perpendicular to the lines of minimal tension. All wounds were separated by at least 2 cm to minimize adjacent wound effects. Each open wound edge was injected with 100 μL PBS (control) or 2.0 mg/μL FMOD in PBS (100 μL x 2 edges = 200 μL total/wound). Wounds were then marked with permanent dye and primarily closed with 3–0 Nylon mattress sutures. Sutures were removed 2 weeks post-injury, and wounds were harvested at 8 weeks post-injury. Tissues were bisected centrally between the sutures and perpendicular to the long axis of each wound[13,14].

### Histology and immunohistochemistry (IHC) staining

After fixation in 10% neutral-buffered formalin (Thermo Fisher Scientific) for 24 h, skin samples were dehydrated, bisected centrally, paraffin-embedded, and sectioned at 5 μm increments for hematoxylin and eosin (H&E) staining, Masson's trichrome staining, or IHC staining.

Antibodies used for IHC staining include mouse anti-α-SMA antibody [4A4] (GTX60466, GeneTex, Irvine, CA; 1: 500 dilution), rabbit anti-cleaved caspase-3 antibody (#9661, Cell Signaling Technology, Danvers, MA; 1:400 dilution), and horse anti-mouse/rabbit IgG antibody (H + L) (Universal), Biotinylated (BA-1400-2.1, Vector Laboratories, Newark, CA; 1:50 dilution). IHC staining was visualized by VECTASTAIN® ABC-HRP Kit, Peroxidase (Standard) (PK-4000; Vector Laboratories). The number of IHC-positively stained cells and nuclei across the entire wound area was counted under a microscope from two centrally bisected sections of each wound sample. The ratio of IHC-positively stained cells to the total number of cells (indicated by the number of nuclei) was then calculated to determine the density of α−SMA[+] or cleaved caspase-3[+] cells.

For Picrosirius red (PSR) staining, 10 μm increments were used as previously described[13,14,19,20]. Photographs were examined under an Olympus IX71 microscope coupled with a DP73 camera and cellSens Standard 1.9 software (Olympus, Cypress, CA).

### RDF isolation, maintenance, and fibroblast-to-myofibroblast conversion

Adult RDFs were isolated from the dorsal skin of adult male Sprague-Dawley rats and maintained in Dulbecco's modified Eagle's medium (DMEM; Thermo Fisher Scientific) supplemented with penicillin/streptomycin (1% v/v; Thermo Fisher Scientific) and fetal bovine serum (FBS; 10% v/v; Thermo Fisher Scientific) as previously described[66]. RDFs were tested negatively for mycoplasma contamination by the Universal Mycoplasma Detection Kit (ATCC). RDFs at passage 3 were used for all in vitro tests.

2.5 ng/mL TGFβ1 (MilliporeSigma, Burlington MA) was added to sub-confluent RDFs daily for 4 days to induce fibroblast-to-myofibroblast conversion, and the converted myofibroblasts were confirmed by significantly increased α-SMA expression before the subsequent analyses.

## Human dermal fibroblast maintenance and fibroblast-to-myofibroblast conversion

Human newborn foreskin BJ fibroblasts (CRL-2522™; ATCC) at passage 27 were used for genomic editing and RNAi, as well as fibroblast-to-myofibroblast conversion and apoptosis analysis. Commercially available primary NHDFs, FB-AA36 (PCS-201-012; ATCC), FB-C31, and FB-C51 (CC-2511; Lonza, Walkersville, MD) were used at passage 3 (Supplementary Table 1) for fibroblast-to-myofibroblast conversion and apoptosis analysis. These cells were maintained in FGM™ Fibroblast Growth Medium BulletKit™ (Lonza). Subsequently, 5.0 ng/mL TGFβ1 was added to sub-confluent human fibroblasts daily for 6 days to induce fibroblast-to-myofibroblast conversion, and the converted myofibroblasts were confirmed by significantly increased α-SMA expression before the subsequent analyses. Lower concentration (2.5 ng/mL) and a short time (3 days) of TGFβ1 inducing was only used for comparing the fibroblast-to-myofibroblast conversion ability of FB-AA36 and its race, gender, and age-matched human keloid fibroblast KB-AA35 (CRL-1762; ATCC; Supplementary Table 1).

Meanwhile, immediately following aseptic surgical removal, the human skin samples were shipped in sterile containers on ice by NDRI. Upon acceptance, human dermal cells were isolated following the established protocol[67] with minor modifications. Briefly, human samples were transferred into a cell culture dish and cut into small pieces approximately 4 mm in diameter. The small pieces of human skin were incubated with 5 U/mL dispase (STEMCELL Technology, Vancouver, BC, Canada) in a 50 mL tube at 4 °C overnight before separating the dermis from the epidermis. Then, the dermis was transferred to a new 50 mL tube and incubated with 1 mg/mL collagenase type IV (STEMCELL Technology) for 1 h at 37 °C, with shaking every 15 min. Then, isolated cells were passed through a 100 μm strainer (Corning Inc., Corning, NY) and collected by centrifuge. The resident dermis was returned to the 50 mL tube and treated with fresh collagenase type IV for another 1 h. Finally, cells collected from three collagenase treatment-cell collection cycles were mixed together and cultured in FGM™ Fibroblast Growth Medium BulletKit™ (Lonza) in three 75 mL flasks till 80% confluence for further investigation.

## Generation of *IL1B-knockout* BJ fibroblasts

Compared to other NHDFs, BJ fibroblasts have a long lifespan with a normal diploid karyotype before senescence and have been broadly used for genomic editing[68–70]. Thus, BJ fibroblasts were chosen as the starting material for *IL1B-knockout* with CRISPR/Cas9 platform technology[30,31]. Passage 27 BJ fibroblasts were used for *IL1B-knockout* generation via the CRISPR-U™ editing approach, conducted by Ubigene Biosciences (Guangzhou, China). The gRNAs 5′-AGC TGGA TGC CGC CAT CCA GAG G-3′ and 5′-AGG TGC TCA GGT CAT TCT CCT GG-3′, and the lentivirus vectors for gRNA (YKO-LV001-dual-gRNA) and Cas9 (YCas-LV002) expression were also constructed by Ubigene Biosciences. The two lentivirus vectors were transfected into BJ cells at MOI 40. The knockout cells were selected by 1.5 μg/mL puromycin (MilliporeSigma) and confirmed by PCR with primers: forward: 5′-TGA GTA CAC ACT TAA CCT CCT TG-3′; reverse: 5′-CCA CAG GGA GGT TAC GAA CC-3′.

In addition, IL1β protein expressed by the *IL1B-knockout* BJ fibroblasts was compared with the WT counterparts with or without lipopolysaccharide (LPS) stimulation. Briefly, $5 \times 10^5$ cells were seeded in 60-mm cell culture plates (Corning Inc.) with or without 100 μg/mL LPS (MilliporeSigma) for 18 h. 35 μg whole-cell proteins extracted by RIPA Lysis buffer (Beyotime Institute of Biotechnology, Haimen, China) in the presence of a protease inhibitor cocktail (Pierce, Rockford, IL) were used for Western blotting, which was visualized with Pierce™ ECL Western Blotting Substrate (Thermo Fisher Scientific)[65]. Antibodies used for Western blotting include goat anti-human IL-1 beta/IL-1F2 antibody (AF-201-NA, R&D systems, Minneapolis, MN; 0.1 μg/mL), rabbit anti-glyceraldehyde-3-phosphate dehydrogenase

(GAPDH) antibody (PA1-987, Thermo Fisher Scientific; 1:3000 dilution), donkey anti-goat IgG (H + L) secondary antibody, HRP (A15999, Thermo Fisher Scientific; 1:5000 dilution), and donkey anti-rabbit IgG (H + L) secondary antibody, HRP (A16023, Thermo Fisher Scientific; 1:5000 dilution).

## Inhibitors

10 μM rhein (Abcam, Waltham, MA)[34] and $6 \times 10^{-5}$ μM IL-1Ra (ProSpec LLc, Mount Vernon, NY)[35] were used to block IL1β signal transduction, respectively.

## RNAi

Following the manufacturer's instructions, predesigned Invitrogen Silencer® siRNAs targeting *IL1B*, *IL1R1*, or *IL1RAP* (Thermo Fisher Scientific) were used to knock down the expression of the respective target genes. The Silencer® Select Negative Control No. 1 siRNA (Thermo Fisher Scientific) was used as the non-targeting negative control siRNA. Forty-eight hours post-transfection, BJ fibroblast-converted myofibroblasts were used for further apoptosis analyses.

## Immunofluorescence (IF) staining

Cells were seeded at a density of $1 \times 10^4$ cells per well on 4-well Lab-TEK® II chamber slides (MilliporeSigma) for IF staining as described previously[13,65]. Mouse anti-α-SMA antibody [4A4] (GTX60466, Gene-Tex; 1: 500 dilution) and goat anti-mouse IgG (H + L) highly cross-adsorbed secondary antibody, Alexa Fluor™ 594 (A-11032, Thermo Fisher Scientific; 1:400 dilution) were used. In addition, 4′,6-diamino-2-phenylindole (DAPI; MilliporeSigma) was used for nuclear counterstaining.

## Quantitative RT-PCR assay

Cells were seeded at a density of $1 \times 10^5$ cells per dish on 10 cm tissue culture dishes. RNA was extracted using the RNeasy® Mini Kit (Qiagen, Valencia, CA) with DNase (Qiagen) treatment to ensure that the samples were not contaminated with genomic DNA. RNA purity was assessed by the Epoch Microplate Spectrophotometer coupled with Gen5 software (version 2.04.11; BioTek Instruments Inc., Winooski, VT). For qRT-PCR, 100 ng RNA was used for a reverse transcriptase reaction with iScript™ Reverse Transcription Supermix for RT-qPCR (Bio-Rad, Hercules, CA). 2 μL RT product was used for real-time PCR with SsoAdvanced™ Universal Probes Supermix (Bio-Rad) and TaqMan® primers/probe sets (Thermo Fisher Scientific) on a QuantStudio3 system (Thermo Fisher Scientific). Concomitant *GAPDH* was performed in separate tubes as a housekeeping standard. Relative gene expression was analyzed with the $_{\Delta\Delta}C_T$ method[65]. TaqMan® primers/probe sets used: Hs00426835_g1 (human *ACTA2*), Hs01555410_m1 (human *IL1B*), and Hs02786624_g1 (human *GAPDH*).

## ELISA for IL1β quantitation

$1 \times 10^6$ cells per well were seeded overnight in 6-well flat-bottom cell culture plates (Corning Inc.). After 16 h of serum starvation, cells were incubated with a 1 mL treatment medium for 48 h. Then, 50 μL of culture supernatant was used for IL1β ELISA, which was performed using the commercially available ELISA MAX™ Deluxe Set Human IL-1β kit (BioLegend, San Diego, CA) per the manufacturer's instruction. IL1β activity was measured by the Epoch Microplate Spectrophotometer at 450 nm, subtracted by readings at 570 nm. IL1β concentrations were then calculated against a standard curve.

## Cell apoptosis induction

After serum starvation for 16 h, cells were treated with 2 ng/mL IL1β ± 2.5 ng/mL (100 pM) TGFβ1 ± 12 μg/mL (200 nM) FMOD for 48 h before apoptosis assessment. Recombinant rat IL1β (MilliporeSigma) and human IL1β (PeproTech, Inc.; East Windsor, NJ) were used for rat and human cells, respectively. A 48 hour incubation period was selected to

minimize the potential overlap between the initial apoptotic or pyroptotic response and the subsequent apoptosis-induced pyroptosis.

## TUNEL analysis

Cells were seeded at a density of $1 \times 10^4$ cells per well on 4-well chamber slides (Millicell® EZ Slide; MilliporeSigma) overnight, followed by 16 h serum starvation. TUNEL analysis was conducted 48 h post-treatment employing the In situ Cell Death Detection Kit, Fluorescein (Roche Applied Science, Penzberg, Germany) per the manufacturer's instruction. When the TUNEL assay was used for apoptotic cell quantification, the number of TUNEL-positively stained cells among 1000 cells (determined by DAPI-stained nuclei) was counted under a microscope for each cell culture. The percentage of apoptotic cells was then calculated based on this count.

## Flow cytometry

Generally, cell apoptosis was quantified by using the Dead Cell Apoptosis Kit with Annexin V-fluorescein isothiocyanate (FITC) and PI (Thermo Fisher Scientific) according to the manufacturer's instructions. Briefly, the testing cells were harvested 48 h post-treatment and washed with cold PBS. Cell density was adjusted to $1 \times 10^5$ cells/mL in 1× annexin-binding buffer and stained with Annexin V-FITC and PI for 15 min at room temperature. The apoptosis rate of staining cells was determined using Flow cytometry (BD™ LSRII; BD Biosciences, San Jose, CA) with excitation at 488 nm and emission at 523 nm. A standard protocol was employed to set the gating by using cells treated with staurosporine (50 nM, 2 h; MilliporeSigma) treatment as the apoptosis control[71], and cells treated with LPS (100 ng/mL) for 3 h followed by 1 h co-treatment with LPS and nigericin (5 μm; MilliporeSigma) as the pyroptosis control[72]. After using unstained controls to identify autofluorescence, single-stained controls for PI and Annexin V-FITC were used to set the compensation matrix, and the boundary between the apoptotic and pyroptotic cells was confirmed by comparing the double-stained controls following the instruction of UCLA Broad Stem Cell Research Center Flow Cytometry Core Resource. Using this Annexin V/PI apoptotic detecting method, flow cytometry categorizes the Annexin V−/PI+ population as necrotic cells, the Annexin V+/PI+ population as pyroptotic cells, and the Annexin V+/PI− population as apoptotic cells. Data was analyzed with FCS Express 4 software (De Novo Software, Glendale, CA).

Since an enhanced green fluorescent protein (EGFP) expression cassette was introduced into the *IL1B-knockout* BJ fibroblasts by the lentivirus vector YKO-LV001-dual-gRNA, Annexin V allophycocyanin (APC) conjugate (Thermo Fisher Scientific) and DAPI were used instead to assess the apoptosis of *IL1B-knockout* myofibroblasts. The flow cytometry was conducted with excitation at 633 nm and emission at 660 nm. A similar gating setting procedure described above has also been used for the DAPI and Annexin V-APC staining. Notably, using the resulting DAPI/APC gating, control- and FMOD-treated wild-type BJ-myofibroblasts without *IL1B-knockout* exhibited comparable percentages of apoptotic cells as using PI/FITC gating. Thus, this DAPI/APC gating was used to analyze *IL1B-knockout* BJ-myofibroblasts in the same experimental setting to ensure accurate gating and rigorous and reliable comparison.

## Pull-down assay

Pull-down assay was conducted with Pierce™ His Protein Interaction Pull-Down Kit (Pierce) following the manufacturer's instruction. 150 μg recombinant human FMOD with C-terminal His-tag[15] was immobilized as the bait. His-tag-free 100 μg recombinant human IL1β (SDS-PAGE: 17 kDa, carrier-free; 201-LB-010/CF, R&D Systems), 100 μg IL1R1 (SDS-PAGE: 55 kDa, carrier-free; 269-1R-100/CF, R&D Systems), or 100 μg IL1RAP (SDS-PAGE: 70−75 kDa, carrier-free; 676-CP-100/CF, R&D Systems) were used as prey, respectively to assay their interaction with FMOD separately. Moreover, 100 μg IL1β, 50 μg IL1R1, and 50 μg

IL1RAP were incubated together in vitro at 4 °C overnight for IL1β-IL1R1-IL1RAP complex formation before being used as the prey to examine the binding between the ternary complex with FMOD. Furthermore, 200 μg whole membrane protein from BJ-myofibroblasts were extracted by Mem-PER™ Plus Membrane Protein Extraction Kit (Thermo Fisher Scientific) and incubated with FMOD in vitro to confirm the binding of IL1β-IL1R1-IL1RAP ternary complex with FMOD in biological situations. The elution was examined by SDS-PAGE and Western blotting, as previously described[73]. Antibodies used for pull-down assay include rabbit anti-fibromodulin antibody (H50) (sc-33772, Santa Cruz Biotechnology, Santa Cruz, CA; 1:500 dilution), goat anti-human IL-1 beta/IL-1F2 antibody (AF-201-NA, R&D systems; 0.1 μg/mL), rabbit anti-IL1 receptor I/IL-1R-1 antibody (ab154524; Abcam; 1:1000 dilution), mouse-anti IL-1RAcP (D5) (sc-376872, Santa Cruz Biotechnology; 1:100 dilution), donkey anti-goat IgG (H + L) secondary antibody, HRP (A15999, Thermo Fisher Scientific; 1:5000 dilution), donkey anti-rabbit IgG (H + L) secondary antibody, HRP (A16023, Thermo Fisher Scientific; 1:5000 dilution), and donkey anti-mouse IgG (H + L) cross-adsorbed secondary antibody, HRP (SA1-100, Thermo Fisher Scientific; 1: 2000 dilution).

## SPR analysis

Bindings between FMOD with IL1R1 and IL1RAP were further characterized on a Biacore 3000 instrument (Biacore AB, Uppsala, Sweden) by the UCLA SPR Core as described before[73]. Briefly, recombinant human FMOD was immobilized on CM5 sensor chips (GE Healthcare) by amine coupling, while IL1R1 and IL1RAP were dissolved in a pH 7.4 HBS-EP buffer containing 0.15 M NaCl, 10 mM HEPES, 3 mM EDTA, and 0.005% polysorbate 20 (GE Healthcare) as the solution phases, respectively. The solution traversed through the sensors at 50 μL/minute flow rate. Low-retention polypropylene tubes (Corning Inc.) were used throughout. Raw data were processed by subtracting the responses in the reference flow cell and the buffer blanks (double referencing), while affinity characters were calculated by plotting the analysis concentration versus the biosensor response and expressed in resonance unit (RU). Besides, kinetic studies were conducted with Scrubber 2.0 (BioLogic Software Pty Ltd., Campbell, Australia) and interpreted by the $k_m/k_a/k_d$ solution[74].

## In silico protein-protein interaction analysis

The crystal structure of the human IL1β-IL1R1-IL1RAP ternary complex was obtained from Protein Data Bank (PDB; Piscataway, NJ; PDB ID: 4DEP)[49]. Due to the lack of a determined full structure, the 3D structure of human FMOD was generated on the I-TASSER server (Iterative Threading Assembly Refinement; Yang Zhang Laboratory, University of Michigan, Ann Arbor, MI)[75].

In silico protein-protein interaction was predicted via the Patch-Dock server (Institute of Molecular Medicine, Tel Aviv University, Tel Aviv, Israel) and then refined and scored according to the binding energies by Fast Interaction Refinement in molecular Docking (Fire-Dock) in the same server[76,77]. The complex structure with the lowest binding energies was chosen for further analysis. Hydrogen bonds were detected by the PDBePISA server (v1.52; EMBL-EBI; Cambridgeshire, UK), while hydrophobic contacts were determined by the Arpeggio server (University of Cambridge, Cambridge, UK)[78].

## In situ PLA

Direct IL1β-IL1R1 and IL1R1-IL1RAP interactions were visualized by Duolink® in situ PLA assay (MilliporeSigma), respectively, according to the manufacturer's instructions. Briefly, cells were seeded at a density of $1 \times 10^4$ cells per well on 4-well chamber slides (Millicell® EZ Slide; MilliporeSigma) overnight, followed by 16 h serum starvation. Being treated with the fresh medium containing FMOD and/or IL1β, cells were fixed by ice-cold acetone at different timepoints. Blocked with the Duolink® blocking solution at room temperature for 60 min, slides

were incubated with rabbit anti-IL1 receptor I/IL-1R-1 antibody (ab154524, Abcam; 1:200 dilution) and goat anti-human IL-1 beta/IL-1F2 antibody (AF-201-NA; R&D systems; 5 µg/mL) or mouse-anti IL-1RAcP (D5) (sc-376872, Santa Cruz Biotechnology; 1:50 dilution) antibody at 4 °C overnight. Then, the slides were washed and incubated with the Duolink® In Situ PLA® Probe Anti-Rabbit MINUS (for IL1R1) coupled with the Probe Anti-Goat PLUS (for IL1β) or the Probe Anti-Mouse PLUS (for IL1RAP) at 37 °C for 60 min. After washing, slides were incubated with the ligation-ligase solution for 30 min at 37 °C, then the amplification solution for 100 min at 37 °C. Last, slides were incubated with Phalloidin-iFluor 488 reagent (ab176753, Abcam; 1:800 dilution) for 1 h at room temperature for outlining the cells and mounted with Duolink® In Situ Mounting Medium with DAPI. Confocal laser scanning microscopy (CLSM) was conducted on a Leica TCS SP8 Confocal Laser Scanning Platform (Leica Biosystems, Wetzlar, Germany). The total number of PLA signals per cell was counted by Image J (version 1.52q; NIH).

### Flow cytometry of PLA-stained cells

IL1β-IL1R1 and IL1R1-IL1RAP bindings at 24 h post-treatment were also assessed by Duolink™ flowPLA Detection Kit−Green (MilliporeSigma) according to the manufacturer's instructions. Adherent BJ-myofibroblasts were detached using Accutase® (STEMCELL) for surface epitope protection. In addition, 3% fetal bovine serum (Thermo Fisher Scientific) was added to wash buffers to minimize cell loss. The same primary antibodies and probes of the in situ PLA assays described above were used, while cells were resuspended in 1x PBS for flow cytometry. Flow cytometry was performed the same day of the staining.

### Schematic illustrations

Icons and schematic illustrations for the mechanism of action (Supplementary Fig. 17) were created in BioRender. Li, C. (2005). https://BioRender.com/u94o620. (Agreement number: FC280ST9CW).

### Statistics and reproducibility

Animals were randomly assigned to each treatment group. Early termination of animal experiments was based on either significant morbidity for animals or a loss of body weight of more than 10%, in accordance with the approved protocols provided by the Chancellor's Animal Research Committee at UCLA; however, no animals were sacrificed for early termination during this study. All data were collected blindly, while all samples were included in the analysis. The number of biological replicates of each experiment is defined in each figure legend. All statistical analyses were conducted in consultation with the UCLA Statistical Biomathematical Consulting Clinic. Statistical analysis was computed by Prism (version 8.2.1; GraphPad, San Diego, CA). Given the small sample size ($N < 50$), Shapiro-Wilk's method was recruited for the normality test as recommended[79,80]. One-way ANOVA and two-tailed unpaired $t$-test were used for parametric data, while Mann-Whitney $U$ and Kruskal-Wallis ANOVA tests were used for nonparametric data. Details of statistical analysis were mentioned in each Figure legend, and the exact $P$-value was provided. $P < 0.05$ (*) was considered a difference, while $P < 0.005$ (**) was recognized as a statistically significant difference based on the recent suggestion[81].

### Reporting summary

Further information on research design is available in the Nature Portfolio Reporting Summary linked to this article.

## Data availability

*Fmod*[−/−] mice (B6;129-FMOD<tm1Aol >/SooJ) are available at the Jackson Laboratory Repository with the JAX Stock No. 037010. All data generated or analyzed during this study are included in this article and its Supplementary Information. Source Data are provided with this paper.

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

## Acknowledgements

This study was supported by NIH NIDCR (R44DE024692 (CS and ZZ) and SB1DE026972 (KT, CS, and ZZ)), NIH NIAMS (R44AR064126 (CS and ZZ)), China Postdoctoral Science Foundation (2020M683267 (XP)), Natural Science Foundation of Chongqing, China (cstc2020jcyj-bshX0107 (XP)), and Chongqing Medical University Program for Youth Innovation in Future Medicine (W0095 (XP)). *Fmod*⁻ᐟ⁻ mice were originally provided by Dr. Ake Oldberg (Lund University, Sweden). We acknowledge the use of tissue procured by the NDRI with support from NIH grant U42OD11158. Flow cytometry was conducted with the support of UCLA Broad Stem Cell Research Center Flow Cytometry Core Resource. CLSM was performed at the Center for NanoScience Institute Advanced Light Microscopy/Spectroscopy Shared Resource Facility at UCLA with funding support from NIH Shared Instrumentation Grant S10OD025017 and NSF Major Research Instrumentation Grant CHE-0722519. We would also like to acknowledge the assistance of Dr. Randy Levinson and the AI tools, Copilot (copilot.microsoft.com) and Grammarly (app.grammarly.com), in refining the language and enhancing the clarity of this manuscript.

The content is solely the responsibility of the authors and does not necessarily represent the official views of the NIH.

## Author contributions

Conception and design of the work: Zhong Zheng. Data collection: Wenlu Jiang, Xiaoxiao Pang, Pin Ha, Grace Xinlian Chang, Yuxin Zhang, and Zhong Zheng. Data analysis and interpretation: Wenlu Jiang, Xiaoxiao Pang, Chenshuang Li, and Zhong Zheng. Drafting the article: Wenlu Jiang, Xiaoxiao Pang, and Zhong Zheng. Critical revision of the article: Chenshuang Li, Lawrence A. Bossong, Kang, and Chia Soo, and Zhong Zheng. Final approval of the version to be published: Kang Ting, Chia Soo, and Zhong Zheng.

## Competing interests

Drs. Kang Ting, Chia Soo, and Zhong Zheng are inventors of fibromodulin-related patents assigned to UCLA. Drs. Kang Ting, Chia Soo, and Zhong Zheng are founders of Scarless Laboratories Inc. and Saint Therapeutics Inc., which sublicense fibromodulin-related patents from the UC Regents, who also hold equity in the company. Drs. Kang Ting, Chia Soo, and Zhong Zheng are also officers of Scarless Laboratories, Inc. and Saint Therapeutics Inc. The remaining authors declare no competing interests.
