## [Transparent Peer Review file · Nature Communications]

Fibromodulin selectively accelerates myofibroblast apoptosis in cutaneous wounds by enhancing interleukin 1 β signaling

Corresponding Author: Dr Zhong Zheng

Version 0:

Reviewer comments:

Reviewer #1

(Remarks to the Author)

The manuscript of Jiang et al aims to provide mechanistic insight into the action of Fibromodulin on efficient wound healing in adults. By using currently available fibroblast cell lines derived from different organisms, including mouse, rat, and human, the authors have confirmed the ability of Fibromodulin to significantly promote wound healing process which is well aligned with their previous findings. Furthermore, in porcine wound model, Fibromodulin significantly reduced scar size and decreased number of myofibroblast during remodeling stage. This phenomenon led the authors to investigate the role of Fibromodulin on clearance of myofibroblast in the late stage of wound healing. The primary conclusion is that Fibromodulin selectively induces myofibroblast apoptosis and thus accelerates the removal of residual cells upon wound closure. To identify the relationship between Fibromodulin and IL1 β mediated apoptosis process, they performed in silico analyses and revealed that Fibromodulin directly bound IL1 β -IL1R1-ILRAP ternary complex which was confirmed by surface plasmon resonance and pull-down assay.

An important observation that the authors have made is that Fibromodulin have almost no ability to affect resting fibroblast and only induce apoptosis of activated/converted fibroblast, which is well aligned with other published work that overlaps on some points. These findings suggest the therapeutic potential of Fibromodulin to treat fibrotic disease through selective myofibroblast clearance. The results are clear, well controlled, high quality, and internally consistent.

The key conclusion is that mechanism of action underlying Fibromodulin induced apoptosis is the stabilization of IL1 β -IL1R1-ILRAP ternary complex by Fibromodulin.

Specific comments:

1. The fibroblast cell lines driven from animals and patients are well described and appropriately chosen. The authors described the genders and ages of the donors adequately. I have no concerns about usage of the cell lines.
2. Except for the experiments using knockout BJ fibroblasts, Annexin V and Propidium Iodide were used to monitor apoptosis. Quadrant of some plots (e.g., Fig 4C) should be re-arranged so that apoptotic population is clearly distinguished from non-apoptotic population. In addition, there are no PI positive population in Fig 2E. Is it simply because of the short incubation time which was not enough to observe the late apoptosis or technical issues?
3. In their previous paper (PMID: 29201497), the authors reported Fibromodulin treatment significantly reduces the mRNA level of TGF β 1-related genes during adult rat cutaneous wound healing. In the line with current study, the authors consider this phenomenon is also result from enhanced IL1 β pathway driven by Fibromodulin? More discussion integrating previous finding and current model will improve the impact of the study.
4. Continuously, the pro-apoptotic effect of Fibromodulin was not rescued by TGF β 1, while the effect of IL1 β was almost completely blocked. If function of Fibromodulin is stabilizing IL1 β -IL1R1-ILRAP ternary complex, TGF β 1 should also affect the Fibromodulin-induced apoptosis. Enhanced stability of the ternary complex seems not enough to explain this difference. If this is the case, simply treating excessive amount of IL1 β may work as efficient as Fibromodulin treatment. Although described in the discussion session, this should be more clearly addressed.
5. Description of Fig 4C-E in main text is not consistent with the data (Line 242), presumably due to simple mistake. Again, the quadrant of plot needs rearrangement.
6. Based on proposed model, Fibromodulin can directly bind to IL1R1 and ILRAP. The overall results from the experiments

using cell lines derived from different organism suggest that the amino acids serving binding interface are conserved among vertebrates, at least among mammals. If the authors could find some similarity of amino acid sequences among species through sequence alignment, this will strongly support the model. Additional further experiments using mutant Fibromodulin (identified by alignment and structural comparison) may further strengthen the conclusion of this study. If not applicable, the authors may find some antibodies against Fibromodulin, IL1R1, or ILRAP that can interfere with Fibromodulin binding to IL1 β -IL1R1-ILRAP ternary complex. Competition assay using these antibodies may be sufficient to prove the proposed mechanism of action.

Reviewer #2

(Remarks to the Author)

In this manuscript, the authors claim that the matricellular proteoglycan fibromodulin (FMOD) reduces scarring by inducing apoptosis of myofibroblasts in an IL1 β -dependent manner. While most of the experiments are conclusive, the mechanism by which FMOD induces apoptosis remains unclear. Additional experiments and clarifications are needed to gain a clear understanding. Please see my specific comments below:

1. If the authors believe that FMOD kills myofibroblasts to reduce scarring, they should stain for cleaved caspase-3 in their rat and porcine wound models.
2. The mechanism by which FMOD kills myofibroblasts in an IL1 β -dependent manner is unclear. The authors provided only speculative discussion about the potential roles of IL1 β and TGF β in the expression of BCL2 family members. These speculations should be verified by examining the levels of BCL2 family proteins, which could help understand why TGF β blocks IL1 β -induced apoptosis but does not block FMOD-induced apoptosis, and how IL1 β induces apoptosis.
3. Why are the cells serum-starved before treatment with FMOD, TGF β , and IL1 β ? Serum starvation could make them more susceptible to any treatment. Have the authors starved the control cells too? This should be mentioned clearly in the figure legend.
4. The reduction of IL1 β induced by the caspase-1 inhibitor suggests that caspase-1 is activated by FMOD. To my knowledge, caspase-1 is activated by the inflammasome. The authors should check if FMOD activates the inflammasome by looking for inflammasome activation markers (e.g. caspase-1 activation, GSDMD cleavage...). The authors should determine if the observed cell death induced by FMOD is pyroptosis and/or apoptosis.
5. In line 242, the authors wrote, "Concurrently, FMOD-mediated apoptosis was significantly greater in the IL1B, IL1R1, and IL1RAP knockdown BJ-myofibroblasts." Where do the authors see this? This is the opposite because there is ~6% of cell death induced by FMOD in the non-targeting siRNA (FMOD-ctl) and only ~1-2% in the IL1B, IL1R1, and IL1RAP knockdown. In any case, 6% of apoptosis is quite low to draw meaningful conclusions about the effect of IL1B, IL1R1, and IL1RAP knockdown.
6. In Fig 5, the IP figures should be better annotated, e.g., His-tagged FMOD IP. The figure legend is also light. The authors should clearly state when it is an in vitro incubation of recombinant His-tagged FMOD with recombinant IL1 β , IL1R, and IL1RAP.

Reviewer #3

(Remarks to the Author)

In this manuscript, the authors investigate the role of FMOD in inducing myofibroblast apoptosis after wound closure to reduce scarring in small and large animal models. Mechanistically, FMOD enhances the formation of the IL1 -IL1R1-ILRAP ternary complex to increase the apoptosis of myofibroblasts and keloid- and hypertrophic scar-derived cells. The authors propose that persistence of myofibroblasts during tissue regeneration is a key cause of fibrosis in most organs and that FMOD represents a promising, broad-spectrum anti-fibrotic therapeutic.

Overall the manuscript is very well written with minimal typographical mistakes. It contains a very large amount of high quality data in 6 multipart figures, 10 extended figures and 9 supplementary figures.

Novel aspects of the manuscript:

Fibromodulin, as a leucine-rich small PG is a novel molecule in the extracellular matrix similar to decorin which this group has studied for many years.

The authors have used 3 in vivo models, mouse, rat and red duroc pig, to demonstrate the antifibrotic effects of FMOD including a novel FMOD $^{-/-}$ knockout mouse.

They employed multiple fibroblast strains, including keloid and hypertrophic scar derived strains and strains from different body locations where tension in the skin varies to establish that FMOD induces apoptosis in myofibroblasts, which are considered fundamental in cause and propagation of fibrosis in the skin and other tissues but not in normal skin fibroblasts. They demonstrated that knockdown of IL1B, IL1 receptor1 type I (IL1R1), or IL receptor accessory protein (IL1RAP) expression weakens the effects of FMOD on myofibroblast IL1 β expression and apoptosis.

Using unique pull down assays they demonstrated that FMOD binds to the ILb-IL1R1-IL1RAP ternary complex. In silico analysis of protein-protein interactions between FMOD with IL1 β -IL1R1 binary complex was performed and binding properties and amino acid residue interactions demonstrated.

Finally using in situ IL1R1:IL1RAP-proximity ligation assays, they present evidence that FMOD promotes the direct binding of IL1R1 and IL1RAP on the surface of BJ-myofibroblasts.

The data presented in each of the presented figures is clean, well prepared and very interesting. Their results are novel, original, and important to the field of fibrosis and its treatment.

Review of the methods provides enough information for the experiments to be reproducible. The standards of methodology is outstanding both in the number and quantity of the experiments and the range of methods employed.

Concerns with the manuscript:

Their conclusions and discussion of their findings is justified and presented in the thorough knowledge of the context of the existing literature. In the conclusions the discussion of the mechanisms of IL-1b induced apoptosis draws heavily on data not contained in the manuscript but described in the literature.

The manuscript is a difficult read due to the large volume of data presented and the frequent and heavy dependence on data in the supplementary and extended figures. For example, one large section the results from line 198 - line 231 depends entirely on data presented not in the figures of the paper but in the extended and supplementary figures which may be difficult for readers to switch back and forth to understand and confirm their assertions.

The in vivo models and methods were reviewed and appear to be conducted in a standardized and ethical fashion but it should be noted that excision wounds are not good models of human scarring where deep partial thickness injuries appear to activate fibroblasts in the deep dermis of the skin. It is unclear where the excessive mechanical loading occurs in the pig model where other than the size and orientation of the full thickness skin no other skin tension is exerted. Do the authors have literature to support the details of this approach and the characterization of the resultant wound to convince one that HTS are the outcome of their approach?

Can the authors justify the local injection of FMOD to be acceptable mode of delivery to control scarring, where literature exists to support the systemic origin of myofibroblasts derived from mesenchymal bone marrow stem cells as an important component of human fibrotic diseases. see Qin L, Liu N, Bao CL, Yang DZ, Ma GX, Yi WH, Xiao GZ, Cao HL. Mesenchymal stem cells in fibrotic diseases-the two sides of the same coin. *Acta Pharmacol Sin.* 2023 Feb;44(2):268-287. doi: 10.1038/s41401-022-00952-0. Epub 2022 Jul 27. PMID: 35896695; PMCID: PMC9326421.

Version 1:

Reviewer comments:

Reviewer #1

(Remarks to the Author)

The authors have adequately addressed all the comments in the revised manuscript.

I still believe that performing an additional competition assay using antibodies against Fibromodulin, IL1R1, or ILRAP—antibodies that could interfere with Fibromodulin binding to the IL1 β -IL1R1-ILRAP ternary complex—would greatly strengthen the manuscript's conclusion.

However, if, as the authors mentioned, there is no commercially available antibody suitable for this purpose, this experiment should be left for future studies.

Reviewer #3

(Remarks to the Author)

This submission is a revision of the original manuscript. The response to 3 reviewers is detailed robust and contains additional data. The manuscript has been revised in an acceptable fashion. Overall it is an interesting and high quality submission.

Reviewer #4

(Remarks to the Author)

Although the authors have addressed the concerns pointed out by reviewer #2, I think they have not adequately responded to comments #1, 2, and 4.

Original comment #1: The authors added the data on immunohistochemical staining of cleaved caspase-3 (Fig. 1). However, it is difficult to find the positive cells. Therefore, the author should perform the western blot analysis to detect cleaved caspase-3.

Original comment #2: Reviewer #2 requested to assess the levels of BCL2 family proteins. However, no data on BCL2 family proteins were provided.

Original comment #4: The authors added the data on CASP1 mRNA expression (Rebuttal Fig. 2). Because, however, CASP1 mRNA levels do not reflect the activation levels of caspase-1, the authors should evaluate the cleavage of caspase-1 by western blotting. Similarly, the cleavage of IL-1beta and GSDMD should be evaluated. Regarding the flow cytometry analysis, the gating strategy to distinguish between apoptotic and pyroptotic cells is inappropriate.

Other concerns are the experimental replicates and statistical analysis. Regarding the experimental replicates, the authors should mention technical replicates versus biological replicates. In addition, it is not clear whether the "two-tailed unpaired t-test" is the correct statistical test in most of the data.

Version 2:

Reviewer comments:

Reviewer #4

(Remarks to the Author)

The authors have not responded to this reviewer's comments #1 and 2. This reviewer still has concerns about these issues.

Reviewer #1

The manuscript of Jiang et al aims to provide mechanistic insight into the action of Fibromodulin on efficient wound healing in adults. By using currently available fibroblast cell lines derived from different organisms, including mouse, rat, and human, the authors have confirmed the ability of Fibromodulin to significantly promote wound healing process which is well aligned with their previous findings. Furthermore, in porcine wound model, Fibromodulin significantly reduced scar size and decreased number of myofibroblast during remodeling stage. This phenomenon led the authors to investigate the role of Fibromodulin on clearance of myofibroblast in the late stage of wound healing. The primary conclusion is that Fibromodulin selectively induces myofibroblast apoptosis and thus accelerates the removal of residual cells upon wound closure. To identify the relationship between Fibromodulin and IL1 β mediated apoptosis process, they performed in silico analyses and revealed that Fibromodulin directly bound IL1 β -IL1R1-ILRAP ternary complex which was confirmed by surface plasmon resonance and pull-down assay.

An important observation that the authors have made is that Fibromodulin have almost no ability to affect resting fibroblast and only induce apoptosis of activated/converted fibroblast, which is well aligned with other published work that overlaps on some points. These findings suggest the therapeutic potential of Fibromodulin to treat fibrotic disease through selective myofibroblast clearance. The results are clear, well controlled, high quality, and internally consistent.

The key conclusion is that mechanism of action underlying Fibromodulin induced apoptosis is the stabilization of IL1 β -IL1R1-ILRAP ternary complex by Fibromodulin.

Response: Thank you for noticing the key points of our current manuscript and for the overall positive feedback.

Specific comments:

1. The fibroblast cell lines driven from animals and patients are well described and appropriately chosen. The authors described the genders and ages of the donors adequately. I have no concerns about usage of the cell lines.

Response: We appreciate your concurrence on our choice of cell lines for this study. Your agreement reinforces the validity of our experimental design and methodology.

2. Except for the experiments using knockout BJ fibroblasts, Annexin V and Propidium Iodide were used to monitor apoptosis. Quadrant of some plots (e.g., Fig 4C) should be re-arranged so that apoptotic population is clearly distinguished from non-apoptotic population.

Response: We appreciate the opportunity to elaborate on our experimental design in greater detail. For the flow cytometry, our process began with an unstained control to identify autofluorescence, followed by single-stained controls for Propidium Iodide (PI) and Annexin V, which were used to establish the compensation matrix. We then optimized the gate setting by analyzing the internal controls of BJ-myofibroblasts that had not undergone the RNAi process. The gate setting that resulted in similar apoptotic populations of approximately 10% of control-treated BJ-myofibroblasts (without RNAi) and roughly 20% of FMOD-treated BJ-myofibroblasts (without RNAi) as shown in other studies, including the data presented in **Fig. 2, Extend Data Fig. 8** (now

reassigned as **Supplementary Fig. 11**) and **Extend Data Fig. 9** (now reassigned as **Supplementary Fig. 12**), was then employed to analyze other cells in the same experimental set as those depicted in **Fig. 4C** (now reassigned as **Fig. 7c**). This setting, we believe, ensures precise gating and comparability of results. The related information has been added to the revised manuscript in the **Methods/Flow cytometry** section.

We observed that all myofibroblasts derived BJ fibroblasts undergoing siRNA transfection have a higher percentage of the Annexin V⁺/PI⁺ population (considered pyroptotic cells) than those without RNAi. This observation aligns with recent findings that noncoding RNAs enhance cell pyroptosis (as reviewed in PMID: 36387211, PMID: 36059539, and doi.org/10.1515/oncologie-2023-0045). Besides, the knockdown of IL1 β , IL1R1, and IL1RAP resulted in a higher pyroptotic population in BJ-myofibroblasts than those transfected with non-targeting siRNA control, providing new insights into the relationship between IL1 β signaling and pyroptosis (as reviewed in doi.org/10.3389/fimmu.2023.1128358). Since this was not the primary focus of the current manuscript, we did not include it in the discussion; however, we thank you for the opportunity to address this issue. Moreover, we have taken note of recent studies that demonstrate the ability of several apoptosis-related caspases, such as caspase-3 and -8, to induce pyroptosis by cleaving gasdermin E (GESME; PMID: 28045099, PMID: 30361383, PMID: 33133646, and PMID: 33776057). The tight junction between the apoptotic and pyroptotic populations of BJ-myofibroblasts expressing noncoding RNA could be attributed to the apoptosis-to-pyroptosis shift, which might be a consequence of the RNAi procedure. Importantly, cell pyroptosis was not enhanced by FMOD treatment under the current experimental conditions (**Fig. 2e,g** and **Fig. 7c, f-g**), reinforcing our conclusion that FMOD primarily induces apoptosis, rather than pyroptosis, in myofibroblasts. We have incorporated these discussions into the revised manuscript (**Lines 147-151** and **254-259**).

In addition, there are no PI positive population in Fig 2E. Is it simply because of the short incubation time which was not enough to observe the late apoptosis, or technical issues?

Response: We hypothesize that the observed decrease in the PI-positive population is due to inherent characteristics of wild-type BJ myofibroblasts rather than methodological or technical issues. This hypothesis is substantiated by the detection of a clear PI-positive population in the experiments utilizing identical techniques as **Fig. 2E** (now reassigned as **Fig. 2e**) when adult human fibroblasts or BJ-fibroblasts post-RNAi procedures are used as the starting materials instead of wild-type BJ, as originally illustrated in **Fig. 3C** (now reassigned as **Fig. 5c**) and **Fig. 4C** (reassigned as **Fig. 7c**), respectively.

Considering the potential occurrence of apoptosis-induced pyroptosis during extended *in vitro* cell culture, we opted for a relatively short incubation time in our study to mitigate the overlap between the initial apoptosis directly induced by FMOD and the secondary apoptosis-induced pyroptosis. We have incorporated this discussion into the revised manuscript in the **Methods/Cell apoptosis induction** section.

3. In their previous paper (PMID: 29201497), the authors reported Fibromodulin treatment significantly reduces the mRNA level of TGF β 1-related genes during adult rat cutaneous wound healing. In the line with current study, the authors consider this phenomenon is also result from enhanced IL1 β pathway driven by Fibromodulin? More discussion integrating previous finding and current model will improve the impact of the study.

Response: We thank you for this excellent suggestion. Accordingly, we have incorporated a discussion pertinent to this point in the revised manuscript (**Lines 375-401**).

4. Continuously, the pro-apoptotic effect of Fibromodulin was not rescued by TGF β 1, while the effect of IL1 β was almost completely blocked. If function of Fibromodulin is stabilizing IL1 β -IL1R1-ILRAP ternary complex, TGF β 1 should also affect the Fibromodulin-induced apoptosis. Enhanced stability of the ternary complex seems not enough to explain this difference. If this is the case, simply treating excessive amount of IL1 β may work as efficient as Fibromodulin treatment. Although described in the discussion session, this should be more clearly addressed.

Response: We value your suggestion and find the interaction between FMOD and TGF β 1 to be of significant interest, as it potentially operates at multiple levels. As briefly outlined in the manuscript, TGF β 1 can shield myofibroblasts from apoptosis through several mechanisms. These mechanisms are not yet fully understood, but they predominantly operate through TGF β 1's non-Smad-driven signaling transduction.

In addition to augmenting and prolonging the formulation of IL1 β -IL1R1-ILRAP ternary complex to facilitate myofibroblast apoptosis, FMOD can directly bind to TGF β 1. This binding may globally sequester TGF β 1 from binding with its receptors to initiate downstream signal transduction. As demonstrated in our previous studies, FMOD also orchestrates TGF β 1 signaling in a more complex manner, including selectively prolonging Smad-driven pathways and simultaneously attenuating non-Smad-driven signaling transduction. Both the enhancement of IL1 β -IL1R1-ILRAP ternary complex formation and the diminution of TGF β 1 non-Smad-driven signaling by FMOD contribute to its pro-apoptosis potency targeting myofibroblasts. However, these two mechanisms may operate independently. Additionally, as discussed in the manuscript, FMOD's binding sites to the IL1 β -IL1R1-ILRAP ternary complex are different from binding sites to TGF β 1, suggesting that FMOD may interact with the IL1 β -IL1R1-ILRAP ternary complex and TGF β 1 simultaneously (**Rebuttal Figure 1**). This could explain why TGF β 1 is unable to inhibit FMOD-induced myofibroblast apoptosis, but FMOD can counteract TGF β 1's effect on preventing myofibroblast apoptosis. This discussion has been expanded in the revised manuscript (**Lines 375-391**).

Rebuttal Figure 1. *In silico* prediction of the putative interaction between TGF β 1 and FMOD-IL1 β -IL1R1-IL1RAP complex. (A) Even bound to the IL1 β -IL1R1-IL1RAP ternary complex, FMOD retains its ability to bind directly with TGF β 1. The structure of the TGF β 1 dimer is sourced from the PCSB protein data bank, PDB: 3KFD.

(B) The 3-D structure of the TGF β 1 ternary complex is predicted based on the known structure of TGF β 1-T β R2 (TGF β type II receptor; sourced from the PCSB protein data bank, PDB: 3KFD with T β R2 extracellular fragment only) and the predicted structure of T β R1 (TGF β type I receptor; sourced from the AlphaFold Protein Structure Database, AF-P36897). Despite the formation of the receptor complex, which consists of two T β R2 and two T β R1 molecules symmetrically bound to the TGF β 1 dimer, we have chosen to present only half of the structure (comprising one T β R2 and one T β R1) in this instance. This decision was made to simplify the visual presentation. (C) When TGF β 1 is bound to the FMOD-IL1 β -IL1R1-IL1RAP complex, it becomes spatially inaccessible to its receptor, thereby preventing the initiation of the downstream signaling.

5. Description of Fig 4C-E in main text is not consistent with the data (Line 242), presumably due to simple mistake. Again, the quadrant of plot needs rearrangement.

Response: Thank you! We so appreciate your careful observation in identifying the error. As you correctly pointed out, our research demonstrates that the knockdown of *IL1B*, *IL1R1*, or *IL1RAP* significantly impedes FMOD's ability to induce myofibroblast apoptosis. The revised manuscript has implemented the necessary corrections (**Lines 251-254**). Meanwhile, please refer to our previous response regarding the gate setting for the quadrant arrangement.

6. Based on proposed model, Fibromodulin can directly bind to IL1R1 and ILRAP. The overall results from the experiments using cell lines derived from different organism suggest that the amino acids serving binding interface are conserved among vertebrates, at least among mammals. If the authors could find some similarity of amino acid sequences among species through sequence alignment, this will strongly support the model.

Response: This is an insightful point. Our prior publication (PMID: 36515321; Figure 1) shows that FMOD exhibits high conservation across species. We have incorporated this information into the revised manuscript (**Lines 272-274**).

Additional further experiments using mutant Fibromodulin (identified by alignment and structural comparison) may further strengthen the conclusion of this study.

Response: We fully agree that, in general, using mutants could further substantiate a molecule's mechanism of action.

As depicted in **Supplementary Fig. 8** (now reassigned as **Supplementary Fig. 15**), *in silico* analysis predicts that multiple amino acid residues of FMOD are potentially involved in its binding with IL1R and IL1RAP. Notably, these residues are adjacent to each other at the 3-dimensional level rather than in the primary structure of FMOD. Functional FMOD has a complex yet stable 3-dimensional structure comprising multiple α -helixes and a β -sheet assembled from multiple β -strands (PMID: 36515321; Figure 3). This inherent complexity in FMOD's structure can lead to potentially misleading interpretations in mutagenesis-binding studies, which our current research strategy attempts to avoid. For example, if we were to introduce a substantial mutant on an amino acid residue involved in the interaction between FMOD and the IL1 β -IL1R1-ILRAP ternary complex, but the mutation does not significantly alter FMOD's three-dimensional structure, other unmutated amino acid residues might still maintain the interaction to some degree. In that case, this preservation of the proapoptotic potency of FMOD could lead to an oversight of the importance of the target amino acid residue for FMOD binding with the ternary complex. Therefore, data from such a study cannot be considered a conclusive rebuttal of our current model. On the other hand, introducing mutants that significantly disrupt the three-dimensional structure

of FMOD would likely affect its interaction with other binding molecules, such as collagen or TGF β 1. Consequently, a direct comparison with wild-type FMOD would not be feasible, as the altered three-dimensional structure of the mutants could introduce additional variables into our analysis. Therefore, any observed effects of these mutants on myofibroblast apoptosis could not be convincingly attributed to alterations in the interaction between FMOD and the IL1 β -IL1R1-ILRAP ternary complex. Due to FMOD's complex structure, we believe that mutagenesis-binding studies are less likely to provide additional support to strengthen or refute our proposed mechanism, even with significant investment of additional human and financial resources. Moreover, such studies would considerably delay the dissemination of our primary conclusion that FMOD selectively induces myofibroblast apoptosis, thereby accelerating the removal of residual myofibroblasts upon wound closure and eventually reducing scar formation.

If not applicable, the authors may find some antibodies against Fibromodulin, IL1R1, or ILRAP that can interfere with Fibromodulin binding to IL1 β -IL1R1-ILRAP ternary complex. Competition assay using these antibodies may be sufficient to prove the proposed mechanism of action.

Response: We completely agree with the recommendation for the use of antibodies. Unfortunately, despite ongoing, extensive searches throughout the process of conducting these studies, we could not find any suitable antibodies to specifically inhibit the binding of FMOD with IL1R1 or ILRAP.

To briefly summarize, the amino acid residues involved in FMOD's interaction with IL1R1 and ILRAP are situated in its central region. Specifically, residues Arg220-Pro310 are involved in the interaction with IL1R1, and residues Arg265-Asn312 are involved in the interaction with ILRAP (original **Supplementary Fig. 8**, now reassigned as **Supplementary Fig. 15**). Currently, no antibodies are available that target these specific fragments of FMOD.

Meanwhile, IL1R1 interacts with FMOD *via* its fragment Asn133-Gln189, while ILRAP interacts with FMOD *via* its fragment Lys152-Asn176 (original **Supplementary Fig. 8**, now reassigned as **Supplementary Fig. 15**). These fragments overlap with the fragments that mediate the binding of IL1R1 and ILRAP (IL1R1: Asp120-Arg208 and ILRAP: Gly134-Thr291, respectively; original **Supplementary Fig. 8**, now reassigned as **Supplementary Fig. 15**). Therefore, an antibody that inhibits the interaction of FMOD with IL1R1 or ILRAP, without interfacing with the IL1R1-ILRAP interaction, must selectively target IL1R1 at Asp120-Lys132 or Tyr190-Arg208, or target ILRAP at Gly134-Val151 or Leu177-Thr291. Unfortunately, after extensive searching, we have concluded that such antibodies are not currently available.

We also considered using antibodies that do not specifically target the fragments mentioned above. However, we are concerned that these antibodies could inhibit the binding between IL1R1 and ILRAP themselves, which could significantly confound our results and data interpretation. Because of this, we focused instead on RNAi to allow us to precisely and effectively reduce the binding of IL1R1 and ILRAP, to provide robust evidence for our conclusions. We believe that our current approach using RNAi provides the most clear and conclusive evidence for our findings and has clarified this point in the revised manuscript (**Lines 244-254**).

Reviewer #2

In this manuscript, the authors claim that the matricellular proteoglycan fibromodulin (FMOD) reduces scarring by inducing apoptosis of myofibroblasts in an IL1 β -dependent manner. While most of the experiments are conclusive, the mechanism by which FMOD induces apoptosis remains unclear. Additional experiments and clarifications are needed to gain a clear understanding.

Response: Thank you for the overall positive comments and valuable input, which are instrumental in refining our work. We appreciate your contribution to our study.

Please see my specific comments below:

1. If the authors believe that FMOD kills myofibroblasts to reduce scarring, they should stain for cleaved caspase-3 in their rat and porcine wound models.

Response: We appreciate the reviewer's excellent suggestion. In response, we have performed the recommended staining procedures. The results of these procedures have been incorporated into the revised manuscript and can be found in revised **Fig. 1, Lines 115-117**, and **Line 121**. We believe this additional data further strengthens our study and are grateful for the opportunity to improve our work. Thank you for your valuable contribution to our research.

2. The mechanism by which FMOD kills myofibroblasts in an IL1 β -dependent manner is unclear. The authors provided only speculative discussion about the potential roles of IL1 β and TGF β in the expression of BCL2 family members. These speculations should be verified by examining the levels of BCL2 family proteins, which could help understand why TGF β blocks IL1 β -induced apoptosis but does not block FMOD-induced apoptosis, and how IL1 β induces apoptosis.

Response: We respectfully acknowledge the reviewer's thoughtful suggestion to present our findings and interpretations with the appropriate level of certainty. Accordingly, we have revised the manuscript more tightly based on our results (**Lines 364-391**) with a newly modified schematic illustration of the potential mechanism (**Supplementary Fig. 16**). We hope this adjustment maintains the integrity and depth of our scientific discourse while addressing the concerns about potential speculation.

We entirely concur that the mechanism of IL1 β and TGF β in governing apoptosis, including aspects related to BCL2 family proteins, is an important area of investigation. As we briefly mentioned in our manuscript, TGF β 1 inhibits myofibroblast mitochondrial apoptosis by upregulating pro-survival proteins and suppressing the pro-apoptotic sensitizer BAD *via* non-Smad-driven pathways, thereby blocking the IL1 β -induced apoptosis (**Lines 349-363**). However, we believe this line of inquiry falls outside the scope of our current manuscript, and we plan to address this question systemically in a follow-up study.

Meanwhile, FMOD's role in TGF β 1 signaling is complex as well. FMOD can directly bind to TGF β 1, effectively sequestering TGF β 1 globally and preventing it from binding to its receptors to initiate downstream signal transduction. Moreover, as demonstrated in our previous studies, FMOD orchestrates TGF β 1 signaling in a more intricate manner, selectively prolonging Smad-driven pathways while simultaneously attenuating non-Smad-driven signaling transduction. The enhancement of the IL1 β -IL1R1-IL1RAP ternary complex formation and the diminution of TGF β 1 non-Smad-driven signaling by FMOD may both contribute to its pro-apoptotic potency

targeting myofibroblasts, while these two mechanisms may operate independently. Furthermore, as discussed in the manuscript, FMOD's binding sites to the IL1 β -IL1R1-ILRAP ternary complex are different sites from those bonded to TGF β 1, suggesting that FMOD may interact simultaneously with the IL1 β -IL1R1-ILRAP ternary complex and TGF β 1 simultaneously (**Rebuttal Figure 1**). This could explain why TGF β 1 does not block FMOD-induced myofibroblast apoptosis, but FMOD can counteract TGF β 1's effect on preventing myofibroblast apoptosis. We have expanded this discussion in the revised manuscript (**Lines 366-391 and Supplementary Fig. 16**) to more closely align with our data.

Rebuttal Figure 1. *In silico* prediction of the putative interaction between TGF β 1 and FMOD-IL1 β -IL1R1-IL1RAP complex. (A) Even bound to the IL1 β -IL1R1-IL1RAP ternary complex, FMOD retains its ability to bind directly with TGF β 1. The structure of the TGF β 1 dimer is sourced from the PCSB protein data bank, PDB: 3KFD. (B) The 3-D structure of the TGF β 1 ternary complex is predicted based on the known structure of TGF β 1-T β R2 (TGF β type II receptor; sourced from the PCSB protein data bank, PDB: 3KFD with T β R2 extracellular fragment only) and the predicted structure of T β R1 (TGF β type I receptor; sourced from the AlphaFold Protein Structure Database, AF-P36897). Despite the formation of the receptor complex, which consists of two T β R2 and two T β R1 molecules symmetrically bound to the TGF β 1 dimer, we have chosen to present only half of the structure (comprising one T β R2 and one T β R1) in this instance. This decision was made to simplify the visual presentation. (C) When TGF β 1 is bound to the FMOD-IL1 β -IL1R1-IL1RAP complex, it becomes spatially inaccessible to its receptor, thereby preventing the initiation of the downstream signaling.

3. Why are the cells serum-starved before treatment with FMOD, TGF β , and IL1 β ? Serum starvation could make them more susceptible to any treatment. Have the authors starved the control cells too? This should be mentioned clearly in the figure legend.

Response: We thank the reviewer's careful attention to detail regarding our experimental procedures. To confirm, all cells used in our study, including the control groups, were subjected to serum starvation prior to treatment. This is a crucial step in our protocol to synchronize the cells and minimize variability in response to treatment. Accordingly, we have updated the figure legends in the revised manuscript to explicitly state that all cells underwent serum starvation before treatment. Your suggestions have greatly improved the clarity and flow of our work.

4. The reduction of IL1 β induced by the caspase-1 inhibitor suggests that caspase-1 is activated by FMOD.

Response: We appreciate the reviewer's comments and the opportunity to clarify. In response, as shown in **Rebuttal Fig. 2** below, FMOD does not significantly upregulate myofibroblast *CASP1* (the gene encoding caspase-1) expression. Based on new information in **Rebuttal Fig. 2**, we respectfully suggest that the observed reduction of IL1 β expression and myofibroblast apoptosis induced by the caspase -1 inhibitor, rhein, indicates that an interruption of IL1 β signaling will cancel the effects of FMOD on IL1 β expression and myofibroblast apoptosis (**Supplementary Fig. 16**), rather than imply that FMOD activated caspase-1. Again, we thank you for improving our manuscript.

Rebuttal Figure 2. FMOD does not significantly upregulate *CASP1* expression in BJ fibroblast-derived myofibroblasts (BJ-myofibroblasts). Expression of *CASP1* (the gene encoding caspase-1) in BJ-myofibroblasts was normalized to the *CASP1* level without treatment. $N = 3$. Data presented as mean \pm s.d. P values were determined by two-tailed unpaired t -tests. *N.S.*, not significant, $P > 0.05$; *, $P < 0.05$; **, $P < 0.005$.

To my knowledge, caspase-1 is activated by the inflammasome. The authors should check if FMOD activates the inflammasome by looking for inflammasome activation markers (e.g. caspase-1 activation, GSDMD cleavage...).

Response: We appreciate the reviewer's recognition of the extensive research conducted on FMOD's role in inflammatory responses, as summarized in our previous review article (PMID: 37483637). We concur that the potential involvement of FMOD in inflammasome activation is an intriguing area of investigation that warrants further exploration. However, as the new information in **Rebuttal Fig. 2 demonstrates**, FMOD does not significantly upregulate *CASP1* expression in myofibroblasts. Therefore, we respectfully suggest that this particular line of inquiry extends beyond the scope of our current manuscript, which primarily focuses on the role of FMOD in myofibroblast apoptosis. While we acknowledge the importance of understanding the broader implications of FMOD in inflammation and inflammasome activation, we believe these topics would be more appropriately addressed in a separate study. We thank the reviewer for their insightful comments and look forward to the opportunity to delve into these areas in future research.

The authors should determine if the observed cell death induced by FMOD is pyroptosis and/or apoptosis.

Response: Using the Annexin V/PI apoptotic detecting method, flow cytometry categorizes the Annexin V⁻/PI⁺ population as necrotic cells, the Annexin V⁺/PI⁺ population as pyroptotic cells, and

the Annexin V⁺/PI⁺ population as apoptotic cells. Our data show that the Annexin V⁺/PI⁺ population was not upregulated by FMOD treatment (**Fig. 2e,g** and **Fig. 7c, f-g**), indicating that FMOD primarily induces apoptosis rather than pyroptosis in myofibroblasts (**Lines 147-151** and **254-259**).

While pyroptosis has traditionally been considered as caspase-1-related cell death, recent studies have shown that multiple apoptosis-related caspases, such as caspase-3 and -8, can induce pyroptosis by cleaving GSDME (PMID: 28045099, PMID: 30361383, PMID: 33133646, and PMID: 33776057), indicating the occurrence of an apoptosis-to-pyroptosis shift. Therefore, it is plausible that FMOD-induced apoptosis could lead to secondary pyroptosis. We have added this discussion in the revised manuscript (**Lines 444-450**).

5. In line 242, the authors wrote, "Concurrently, FMOD-mediated apoptosis was significantly greater in the IL1B, IL1R1, and IL1RAP knockdown BJ-myofibroblasts." Where do the authors see this? This is the opposite because there is ~6% of cell death induced by FMOD in the non-targeting siRNA (FMOD-ctl) and only ~1-2% in the IL1B, IL1R1, and IL1RAP knockdown. In any case, 6% of apoptosis is quite low to draw meaningful conclusions about the effect of IL1B, IL1R1, and IL1RAP knockdown.

Response: We appreciate your diligence in reviewing our work and thank you for bringing this to our attention. Indeed, the knockdown of *IL1B*, *IL1R1*, or *IL1RAP* significantly impedes FMOD's ability to induce myofibroblast apoptosis. We acknowledge this oversight and have corrected the revised manuscript, which can be found on **Lines 251-254**.

6. In Fig 5, the IP figures should be better annotated, e.g., His-tagged FMOD IP. The figure legend is also light. The authors should clearly state when it is an in vitro incubation of recombinant His-tagged FMOD with recombinant IL1 β , IL1R, and IL1RAP.

Response: We appreciate your constructive suggestion. Accordingly, we have made the necessary modifications to our manuscript as recommended (now reassigned as **Fig. 8** in the revised manuscript). We believe that these revisions have significantly improved the clarity and coherence of our work, making it more accessible and comprehensible to readers.

Reviewer #3

In this manuscript, the authors investigate the role of FMOD in inducing myofibroblast apoptosis after wound closure to reduce scarring in small and large animal models. Mechanistically, FMOD enhances the formation of the IL1 β -IL1R1-ILRAP ternary complex to increase the apoptosis of myofibroblasts and keloid- and hypertrophic scar-derived cells. The authors propose that persistence of myofibroblasts during tissue regeneration is a key cause of fibrosis in most organs and that FMOD represents a promising, broad-spectrum anti-fibrotic therapeutic.

Overall the manuscript is very well written with minimal typographical mistakes. It contains a very large amount of high quality data in 6 multipart figures, 10 extended figures and 9 supplementary figures.

Novel aspects of the manuscript:

Fibromodulin, as a leucine-rich small PG is a novel molecule in the extracellular matrix similar to decorin which this group has studied for many years.

The authors have used 3 in vivo models, mouse, rat and red duroc pig, to demonstrate the antifibrotic effects of FMOD including a novel FMOD^{-/-} knockout mouse.

They employed multiple fibroblast strains, including keloid and hypertrophic scar derived strains and strains from different body locations where tension in the skin varies to establish that FMOD induces apoptosis in myofibroblasts, which are considered fundamental in cause and propagation of fibrosis in the skin and other tissues but not in normal skin fibroblasts.

They demonstrated that knockdown of IL1B, IL1 receptor1 type I (IL1R1), or IL receptor accessory protein (IL1RAP) expression weakens the effects of FMOD on myofibroblast IL1 b expression and apoptosis.

Using unique pull down assays they demonstrated that FMOD binds to the IL β -IL1R1-IL1RAP ternary complex. In silico analysis of protein-protein interactions between FMOD with IL1 β -IL1R1 binary complex was performed and binding properties and amino acid residue interactions demonstrated.

Finally using in situ IL1R1:IL1RAP-proximity ligation assays, they present evidence that FMOD promotes the direct binding of IL1R1 and IL1RAP on the surface of BJ-myofibroblasts.

The data presented in each of the presented figures is clean, well prepared and very interesting. Their results are novel, original, and important to the field of fibrosis and its treatment.

Review of the methods provides enough information for the experiments to be reproducible. The standards of methodology is outstanding both in the number and quantity of the experiments and the range of methods employed.

Response: We greatly appreciate your positive feedback on our work. It is encouraging to know that our research has been well-received. We are committed to maintaining the high standards of our study. We will continue to strive for excellence in our scientific endeavors. Thank you once again for your supportive comments. They serve as a significant motivation for our team.

Concerns with the manuscript:

Their conclusions and discussion of their findings is justified and presented in the thorough knowledge of the context of the existing literature. In the conclusions the discussion of the mechanisms of IL-1b induced apoptosis draws heavily on data not contained in the manuscript but described in the literature.

Response: We appreciate the reviewer's insightful comments. Indeed, the mechanism of apoptosis

is a complex process modulated by many molecules and warrants ongoing investigation. Our discussion in the manuscript is grounded in the references we have cited, providing a solid basis for our data interpretation. However, we acknowledge the importance of presenting our findings with appropriate certainty. We have revised the manuscript in response to your feedback (**Lines 366-391**). We hope this adjustment maintains the integrity of our scientific findings and meets your expectations, and we look forward to any further suggestions you may have to enhance the quality of our manuscript. Thank you for your valuable contribution to our study.

The manuscript is a difficult read due to the large volume of data presented and the frequent and heavy dependence on data in the supplementary and extended figures. For example, one large section the results from line 198 - line 231 depends entirely on data presented not in the figures of the paper but in the extended and supplementary figures which may be difficult for readers to switch back and forth to understand and confirm their assertions.

Response: We sincerely apologize that the initial submission of our manuscript was difficult to read. In response to your feedback and following the guidelines of *Nature Communications*, we have reorganized the figures in our manuscript to present our data more effectively. We believe these changes significantly improve the readability and flow of our work. We hope these modifications meet your expectations and enhance your understanding of our research. We appreciate your patience and valuable feedback in this matter.

The in vivo models and methods were reviewed and appear to be conducted in a standardized and ethical fashion but it should be noted that excision wounds are not good models of human scarring where deep partial thickness injuries appear to activate fibroblasts in the deep dermis of the skin.

Response: We appreciate your comment and understand the limitations of excision wound models in fully replicating human scarring, particularly in the context of deep partial-thickness injuries. Excision wound models are indeed simplified representations and may not capture all the complexities of human wound healing and scar formation. We know that additional factors involve human wound healing, which can significantly influence the healing outcome and the extent of scarring. For instance, deep partial-thickness burn injuries have more complex inflammatory responses and healing dynamics, including the activation of fibroblasts in the deep dermis of the skin, and thus, are associated with a common complication—hypertrophic scarring. While we acknowledge these limitations, it's important to note that animal models, such as excision wounds, provide valuable insights into the fundamental processes of wound healing and scar formation. They allow for controlled experimental conditions and the ability to manipulate specific variables. However, your point underscores the importance of interpreting the results from such models with caution and the need for further validation in more clinically relevant models or human studies. We appreciate your insightful comment and have added the relative discussion in the revised manuscript (**Lines 429-439**).

It is unclear where the excessive mechanical loading occurs in the pig model where other than the size and orientation of the full thickness skin no other skin tension is exerted. Do the authors have literature to support the details of this approach and the characterization of the resultant wound to convince one that HTS are the outcome of their approach?

Response: We appreciate the opportunity to provide further clarification on our choice of model. Previous studies have indeed demonstrated that the red Duroc pig exhibits scarring patterns that closely resemble human hypertrophic scarring (HTS) or fibroproliferative scarring (PMID: 17727465, PMID: 21606834, PMID: 18211585, PMID: 16905264; PMID: 17438498, PMID: 18354295, PMID: 32637158, among others). A recent systematic review has concluded that the red Duroc pig has a genetic predisposition to form HTS, thereby presenting a suitable model for HTS (PMID: 35529910). Furthermore, the degree of skin fibrosis following excisional wound closure in the red Duroc pig has been correlated with mechanical stress levels, with larger wounds resulting in greater mechanical tension post-closure (PMID: 21606834). To maximize the relevance of our model to human HTS, we created wounds with excessive mechanical loading in red Duroc pigs, as utilized in our previous studies (PMID: 29392829). This approach most closely approximates HTS formation in humans. We hope this additional information addresses your query and provides a clearer understanding of our choice of model.

Can the authors justify the local injection of FMOD to be acceptable mode of delivery to control scarring, where literature exists to support the systemic origin of myofibroblasts derived from mesenchymal bone marrow stem cells as an important component of human fibrotic diseases. see Qin L, Liu N, Bao CL, Yang DZ, Ma GX, Yi WH, Xiao GZ, Cao HL. Mesenchymal stem cells in fibrotic diseases-the two sides of the same coin. *Acta Pharmacol Sin.* 2023 Feb;44(2):268-287. doi: 10.1038/s41401-022-00952-0. Epub 2022 Jul 27. PMID: 35896695; PMCID: PMC9326421.

Response: We appreciate your comments and the opportunity to discuss our methodology and focus further. The local injection has indeed been widely accepted for investigations into wound healing and scarring therapies, leading to the development of several drug candidates (PMID: 30603411, PMID: 32734786, PMID: 19128261, PMID: 18787533, PMID: 30349773, among others).

We acknowledge that dermal myofibroblasts have diverse origins, including dermal fibroblasts and mesenchymal bone marrow stem cells. Despite their phenotypic and functional heterogeneity, common characteristics of dermal injury-activated myofibroblasts include the development of contractile force transmission to the extracellular matrix (ECM) and the promotion of wound defect contraction and closure. Moreover, most dermal injury-activated myofibroblasts are predominantly derived from dermal fibroblasts (PMID: 17299435, PMID: 20962852, PMID: 25395868). Therefore, our current study is primarily focused on dermal fibroblast-derived myofibroblasts. However, the question of whether FMOD also induces apoptosis in myofibroblasts derived from other sources, such as mesenchymal stem cells, is indeed an interesting one that warrants further investigation. We have expanded on this discussion in the revised manuscript (**Lines 439-444**).

Again, we thank you for your insightful comments and welcome any further suggestions you may have to enhance our work.

Reviewer 1:

The authors have adequately addressed all the comments in the revised manuscript.

Response: We appreciate your thorough review and your acknowledgment of the efforts we made to address all the comments.

I still believe that performing an additional competition assay using antibodies against Fibromodulin, IL1R1, or ILRAP—antibodies that could interfere with Fibromodulin binding to the IL1 β -IL1R1-ILRAP ternary complex—would greatly strengthen the manuscript's conclusion. However, if, as the authors mentioned, there is no commercially available antibody suitable for this purpose, this experiment should be left for future studies.

Response: Thank you for your insightful comment. We appreciate your suggestion regarding the additional competition assay. We agree that the antibody competition assay would greatly strengthen our conclusions. Unfortunately, as previously mentioned, the lack of available antibodies suitable for this purpose limits our ability to conduct this experiment at this time. We will certainly consider this valuable suggestion for future studies.

Reviewer 3:

This submission is a revision of the original manuscript. The response to 3 reviewers is detailed robust and contains additional data. The manuscript has been revised in an acceptable fashion. Overall it is an interesting and high quality submission.

Response: Thank you for your kind feedback and comprehensive review. We appreciate your acknowledgment of the detailed responses and supplementary data. We are delighted that you found the manuscript engaging and of high quality. Your support is truly appreciated.

Reviewer 4:

Although the authors have addressed the concerns pointed out by reviewer #2, I think they have not adequately responded to comments #1, 2, and 4.

Response: Thank you for taking over the review process and continuing the evaluation. We sincerely appreciate your thoughtful feedback and acknowledgement of our efforts to address the previous concerns. Your comments are incredibly valuable to us, and we assure you that our commitment to delivering the highest-quality outcome remains steadfast. Please find our responses to each of your points below.

Original comment #1: The authors added the data on immunohistochemical staining of cleaved caspase-3 (Fig. 1). However, it is difficult to find the positive cells. Therefore, the author should perform the western blot analysis to detect cleaved caspase-3.

Response: Thank you for your valuable suggestion. We would like to clarify that the primary goal of Fig. 1 is to demonstrate that FMOD significantly reduces the population of myofibroblasts in adult wounds. This observation led us to investigate whether FMOD induces the apoptosis of myofibroblasts, as apoptosis is the main mechanism for clearing these cells in re-epithelialized wounds (PMID: 27158462, PMID: 34439762, PMID: 31792399).

We sincerely appreciate Reviewer #2's insightful suggestion, which encouraged us to conduct cleaved caspase-3 staining in rat and porcine models, as recommended. While we recognize that cleaved caspase-3 staining is commonly used to detect apoptosis, we found this suggestion particularly intriguing. It is important to note that staining methods can only capture a specific moment in time and fail to encompass the dynamic processes, such as the ongoing temporal and spatial changes myofibroblasts undergo during *in vivo* wound healing, including proliferation, differentiation, apoptosis, and clearance.

In our current study, it is imperative to assess the resistance of myofibroblasts in mature wounds when they have largely been cleared from the wounds. This results in a significantly decreased total cell density compared to the peak period. Consequently, observing less than 10% of cells undergoing programmed cell death (as indicated by cleaved caspase-3 staining; Fig. 1) across all tested groups is entirely justifiable. Our findings suggest that cleaved caspase-3 staining may not always be advantageous for detecting cell apoptosis in specific contexts.

Meanwhile, Western blot is viewed as a semi-quantitative, dual-factor analysis that reflects the combination of (1) the proportion of cells expressing the target protein and (2) the pooled expression level of these cells (PMID: 28974114). It is understood that cells at different stages of apoptosis can show varying levels of cleaved caspase-3 (PMID: 10200555; PMID: 17562483). However, myofibroblasts do not undergo apoptosis synchronously during the dynamic wound-healing process. Additionally, formalin fixation can lead to extensive crosslinking of proteins, resulting in a low yield of protein extraction and affecting the sensitivity of Western blot analysis. Considering that less than 10% of cells are cleaved-caspase-3⁺, and the differences between groups are less than 5%, it is impossible to determine the extent to which the Western blot data could be attributed to differences in the percentage of positively stained cells in tissue samples rather than the staining density of individual cells. Therefore, we respectfully disagree that Western blot is a suitable method for this particular study to provide meaningful data due to its inherent disadvantages.

Original comment #2: Reviewer #2 requested to assess the levels of BCL2 family proteins. However, no data on BCL2 family proteins were provided.

Response: We would first like to revisit the background regarding Reviewer #2's requirement to examine BCL2 family proteins. Reviewer #2 commented on our initial submission, *'The authors provided only speculative discussion about the potential roles of IL1 β and TGF β in the expression of BCL2 family members. These speculations should be verified by examining the levels of BCL2 family proteins, which could help understand why TGF β blocks IL1 β -induced apoptosis but does not block FMOD-induced apoptosis and how IL1 β induces apoptosis.'*

In response to Reviewer #2's comments, we have made significant revisions to enhance our discussion of FMOD's mechanism of action in promoting myofibroblast apoptosis. We have intentionally shifted our focus from the BCL2 family proteins and instead concentrated on our

robust evidence regarding IL1 β -trinary complex formation. Our updated schematic illustration (**Supplementary Fig. 16**) now clearly highlights our central discovery: FMOD selectively induces myofibroblast apoptosis by enhancing IL1 β signaling. We believe these revisions adequately address Reviewer #2's concerns about potential speculation. In this updated version, we further clarify that BCL2 family proteins are not the focus of our current investigation. All statements regarding BCL2 family proteins have been removed from **Lines 340-347**, which now concentrate on the primary findings of the current study, to **Lines 429-448**, which discuss the limitations of the current study and suggest directions for future investigation. Therefore, we do not consider it essential to test BCL2 proteins in this study.

We agree that understanding the mechanisms of IL1 β and TGF β in regulating apoptosis, and how FMOD modulates these—especially concerning BCL2 family proteins—represents a promising direction for future investigation (**Lines 429-448**). However, it is essential to acknowledge that this mechanism remains largely unknown, and existing data demonstrate its complexity (PMID: 31792399). Fully unraveling this mystery, which has puzzled researchers for decades, is not feasible in this initial report. While pursuing this line of inquiry is valuable, it falls well outside the scope of our current manuscript. Exploring these complexities would significantly delay the dissemination of our primary conclusion: FMOD selectively induces myofibroblast apoptosis, accelerating the removal of residual myofibroblasts during wound closure and ultimately reducing scar formation. Instead, we intend to systematically address these important questions in a series of future studies. We hope the reviewer understands and agrees with our perspective.

Original comment #4: The authors added the data on CASP1 mRNA expression (Rebuttal Fig. 2). Because, however, CASP1 mRNA levels do not reflect the activation levels of caspase-1, the authors should evaluate the cleavage of caspase-1 by western blotting. Similarly, the cleavage of IL-1beta and GSDMD should be evaluated.

We sincerely appreciate the excellent suggestion made by the reviewer. Following this recommendation, we tested the expression and cleavage of caspase-1 using Western blotting (**Rebuttal Fig. 2_1**). In agreement with the previous transcriptional analysis, we found no significant enhancement in caspase-1 expression or cleavage in response to FMOD treatment.

Rebuttal Fig.2_1. FMOD does not significantly upregulate caspase-1 expression or cleavage in BJ fibroblast-derived myofibroblasts (BJ-myofibroblasts). Expression and cleavage of caspase-1 were detected by Western blotting with Cell Signaling Technology antibody #3866. 30 μ g total protein was loaded for each sample. GAPDH was the housekeeping standard detected with Cell Signaling Technology antibody #97166. Pos., positive control for antibody testing: 1 x 10⁵ cells were seeded into a 6-well plate. After 16 h of serum starvation, cells were treated with 10 μ g/mL lipopolysaccharide for 12 h, then 5 mM ATP before protein collection. P.L., protein ladder. Bands on

Western blots were quantified in relative densitometry units and normalized to the levels without treatment (Control). $N = 3$. Data presented as mean \pm s.d. P values were determined by two-tailed unpaired t -tests. *N.S.*, not significant, $P > 0.05$; *, $P < 0.05$.

Furthermore, FMOD does not upregulate gasdermin D expression, while cleaved gasdermin D was hardly detected by Western blotting, even when the protein loading amount was significantly increased to 80 μ g total protein/lane (**Rebuttal Fig. 2_2**). Therefore, we respectfully argue that this investigation into inflammasome activation exceeds the scope of our current manuscript, which primarily focuses on FMOD's role in myofibroblast apoptosis.

Rebuttal Fig.2_2. FMOD does not significantly altered gasdermin D expression or cleavage in BJ fibroblast-derived myofibroblasts (BJ-myofibroblasts). Full-length gasdermin D was detected by Western blotting with Cell Signaling Technology antibody #39454, while cleaved gasdermin D was detected by Cell Signaling Technology antibody #36425. 80 μ g total protein was loaded for each sample. GAPDH was the housekeeping standard detected with Cell Signaling Technology antibody #97166. Pos., positive control for antibody testing: 1×10^5 cells were seeded into a 6-well plate. After 16 h of serum starvation, cells were treated with 10 μ g/mL lipopolysaccharide for 12 h, then 5 mM ATP before protein collection. P.L., protein ladder. Bands on Western blots were quantified in relative densitometry units and normalized to the levels without treatment (Control). $N = 3$. Data presented as mean \pm s.d. P values were determined by two-tailed unpaired t -tests. *N.S.*, not significant, $P > 0.05$; *, $P < 0.05$.

On the other hand, we have evaluated the effects of FMOD on myofibroblast IL-1 β maturation using ELISA, a more sensitive and quantitative method than Western blot. The relative data are presented in the manuscript on **Lines 208-216** and in **Supplementary Fig. 8**.

Regarding the flow cytometry analysis, the gating strategy to distinguish between apoptotic and pyroptotic cells is inappropriate.

Response: Thank you for giving us the opportunity to further clarify the protocol we used for cytometry analysis. We employed a standard protocol to optimize the gate setting by using cells treated with staurosporine (50 nM, 2 h; MilliporeSigma) as the apoptosis control (PMID: 18638274) and cells treated with lipopolysaccharide (LPS, 100 ng/mL; MilliporeSigma) for 3 h, followed by a 1 h co-treatment with LPS and nigericin (5 μ M; MilliporeSigma) as the pyroptosis control (PMID: 30028908). Unstained controls were used to identify autofluorescence, while single-stained controls for propidium iodide (PI) and Annexin V-fluorescein isothiocyanate (FITC) were utilized to establish the compensation matrix. This approach confirmed the boundary between the apoptotic and pyroptotic cells by comparing the double-stained controls. We also consulted with the UCLA Broad Stem Cell Research Center Flow Cytometry Core Resource, a reputable institution in the field, which validated this strategy as appropriate.

To investigate the cells that underwent *IL1B-knockout*, we applied a similar gating setting procedure for the DAPI and Annexin V-allophycocyanin (APC) staining, following the instruction of UCLA Broad Stem Cell Research Center Flow Cytometry Core Resource. Notably, using the resulting DAPI/APC gating, control- and FMOD-treated wild-type BJ-myofibroblasts without *IL1B-knockout* exhibited comparable percentages of apoptotic cells as those using PI/FITC staining and gating, ensuring a rigorous and reliable comparison.

We have revised the ‘**Flow cytometry**’ section in the ‘**Methods**’ to detail the gating strategy, and we hope this clarifies any confusion and confirms its appropriacy.

Other concerns are the experimental replicates and statistical analysis. Regarding the experimental replicates, the authors should mention technical replicates versus biological replicates.

Response: Thank you for your valuable guidance. In the revised version, we clarified that the N numbers mentioned refer to biological replicates in the ‘**Methods/Statistical analysis**’ section.

In addition, it is not clear whether the “two-tailed unpaired t-test” is the correct statistical test in most of the data.

Response: Thank you for raising this question. A two-tailed unpaired *t*-test compares the means of two independent groups to determine if they differ significantly in either direction (either greater than or less than). The requirements for using the two-tailed unpaired *t*-test include:

- (1) The groups for comparison are independent of each other,
- (2) Data in both groups should follow normal distribution (if this assumption is violated, a non-parametric test should be used instead),
- (3) The variances of the two groups should be roughly equal,
- (4) The dependent variable should be measured on a continuous scale.

All our experimental data met the conditions (1), (2), and (4). Due to the relatively small sample sizes (< 50), we used the Shapiro-Wilk test to assess the normality of the data (PMID: 30648682). Data that passed the Shapiro-Wilk test were treated as parametric and compared using a two-tailed unpaired *t*-test. Conversely, data that failed the Shapiro-Wilk test were analyzed with the non-parametric Mann-Whitney *U* test, as stated in the manuscript. Once again, we thank you for allowing us to clarify this important question. Overall, we are confident that our statistical methods are appropriate.

Response to the reviewer's comments.

Reviewer #4 (Remarks to the Author):

The authors have not responded to this reviewer's comments #1 and 2. This reviewer still has concerns about these issues.

Original comment #1: The authors added the data on immunohistochemical staining of cleaved caspase-3 (Fig. 1). However, it is difficult to find the positive cells. Therefore, the author should perform the western blot analysis to detect cleaved caspase-3.

Response: Thank you for the suggestion.

In the revised Fig. 1, we have improved the quality of the photos by replacing a new microscope bulb, enhancing the visibility of cleaved caspase-3 staining. Furthermore, since fibromodulin (FMOD) effectively induces myofibroblast apoptosis, the overall cell density of FMOD-treated samples was lower than their control. Consequently, the density of cleaved caspase-3+ cells in the images may differ from what is observed with the naked eye. This phenomenon reflects the biological impact of FMOD rather than an imaging artifact.

Meanwhile, Western blot is viewed as a semi-quantitative, dual-factor analysis that reflects the combination of (1) the proportion of cells expressing the target protein and (2) the pooled expression level of these cells (PMID: 28974114). It is understood that cells at different stages of apoptosis can show varying levels of cleaved caspase-3 (PMID: 10200555; PMID: 17562483). However, myofibroblasts do not undergo apoptosis synchronously during the dynamic wound-healing process. Additionally, formalin fixation can lead to extensive crosslinking of proteins, resulting in a low yield of protein extraction and affecting the sensitivity of Western blot analysis. Considering that less than 10% of cells are cleaved-caspase-3⁺, and the differences between groups are less than 5%, it is impossible to determine the extent to which the Western blot data could be attributed to differences in the percentage of positively stained cells in tissue samples rather than the staining density of individual cells. Therefore, we respectfully disagree that Western blot is a suitable method for this particular study to provide meaningful data due to its inherent disadvantages.

Original comment #2: Reviewer #2 requested to assess the levels of BCL2 family proteins. However, no data on BCL2 family proteins were provided.

Response: In Revision 2, we substantially revised our manuscript to provide a more comprehensive discussion of FMOD's mechanism of action in promoting myofibroblast apoptosis. Specifically, we emphasized our robust evidence supporting IL1 β -ternary complex formation as a key driver of this process. We also expanded our discussion regarding the study's limitations and proposed future research directions, including potential mechanisms involving the BCL2 family, to guide subsequent investigations.

We are deeply grateful for the editor's acknowledgment that no further experiments involving BCL2 family members are required at this stage, which reinforces the strength of our primary findings and their alignment with the current scope of the study.